# An analytical study of M$_2$ tidal waves in the Taiwan Strait using an extended Taylor method

Di Wu[1], Guohong Fang[1,2], Xinmei Cui[1,2], Fei Teng[1,2]

[1]The First Institute of Oceanography, State Oceanic Administration, Qingdao, 266061, China

[2]Laboratory for Regional Oceanography and Numerical Modeling, Qingdao National Laboratory for Marine Science and Technology, Qingdao, 266237, China

*Correspondence to*: Guohong Fang (fanggh@fio.org.cn)

**Abstract.** The tides in the Taiwan Strait (TS) feature large semidiurnal lunar (M$_2$) amplitudes. An extended Taylor method is employed in this study to provide an analytical model for the M$_2$ tide in the TS. The strait is idealized as a rectangular basin with a uniform depth, and the Coriolis force and bottom friction are retained in the governing equations. The observed tides at the northern and southern openings are used as open boundary conditions. The obtained analytical solution, which consists of a stronger southward propagating Kelvin wave, a weaker northward propagating Kelvin wave, and two families of Poincaré modes trapped at the northern and southern openings, agrees well with the observations in the strait. The superposition of two Kelvin waves basically represents the observed tidal pattern, including an anti-nodal band in the central strait, and the cross-strait asymmetry (greater amplitudes in the west and smaller in the east) of the anti-nodal band. Inclusion of Poincaré modes further improves the model result in that the cross-strait asymmetry can be better reproduced. To explore the formation mechanism of the northward propagating wave in the TS, three experiments are carried out, including the deep basin south of the strait. The results show that the southward incident wave is reflected to form a northward wave by the abruptly deepened topography south of the strait, but the reflected wave is slightly weaker than the northward wave obtained from the above analytical solution, in which the southern open boundary condition is specified with observations. Inclusion of the forcing at the Luzon Strait strengthens the northward Kelvin wave in the TS, and the forcing is thus of some (but lesser) importance to the M$_2$ tide in the TS.

## 1 Introduction

The Taiwan Strait (TS) is the sole passage connecting the East China Sea (ECS) and the South China Sea (SCS). The strait is approximately 350 km long, 200 km wide and located mostly on the continental shelf with a mean depth of approximately 50 metres. The bottom topography of the TS can be viewed as the extension of the ECS shelf in the north and becomes irregular in the south. The SCS deep basin is located south of the strait and is connected to the Pacific Ocean through the Luzon Strait (LS). An abrupt depth change is present between the TS and the SCS deep basin (Fig. 1).

The tides in the strait feature large M$_2$ amplitudes. The greatest amplitude based on tidal gauge observations along the

western Taiwan coast, reported by Jan et al. (2004b), is 1.73 m at Taichung and is 2.10 m at Matsu near the mainland coast. Matsu is an island located approximately 20 km away from the coast. Satellite observations indicate that the greatest amplitude appears near Haitan Island, located south of Matsu (Fig. 2), and exceeds 2.2 m. Thus, the tidal regime of the $M_2$ constituent has an anti-nodal band near the cross-strait line from Haitan to Taichung, with greater amplitudes in the west and smaller in the east, and this feature is called asymmetry by Yu et al. (2015). Compared to $M_2$, which has maximum amplitude over 2.2 m, the amplitudes of the rest of the constituents are much smaller: the maximum amplitudes of $S_2$, $K_1$ and $O_1$ observed at 11 coastal gauge stations reported by Jan et al. (2004b) are 0.66, 0.39 and 0.27 m, respectively. Figure 2 displays the distribution of the $M_2$ tidal constituent based on the global tidal model DTU10, which is constructed on the basis of multi-mission altimeter observations. Hereafter, we shall regard the DTU10 model results as observations. The tides in the TS have attracted a great number of studies since the 1980s. Yin and Chen (1982) first developed two-dimensional model for tides in the TS without showing tidal currents. Fang et al. (1984) again developed a two-dimensional model and obtained rather accurate distribution of tidal currents. They suggested that the semidiurnal tidal motion in the TS was maintained mainly by the energy flux from the ECS and partly by that from the SCS. Ye et al. (1985) and Lü and Sha (1999) developed three-dimensional models for the strait and also found that the southward energy flux of semidiurnal tides from the ECS was much greater than the northward flux. Lin et al. (2000, 2001) emphasized the anomalous amplification of semidiurnal tides in the strait, and attributed the amplification to a resonance. Jan et al. (2004b) modeled the tides using an optimization approach. Zhu, et al. (2009), Hu et al. (2010), and Zeng et al. (2012) further developed more accurate numerical models. Yu et al. (2017) studied the propagation and dissipation of tidal waves in the strait. It has been well recognized from these numerical investigations that the semidiurnal tides in the TS consist mainly of two oppositely propagating waves, one from north to south and another from south to north. In particular, Fang et al. (1984, 1999) and Ye et al. (1985) suggested that the semidiurnal tidal motion in the TS was maintained mainly by the energy flux from the ECS and partly by that from the SCS. Jan et al. (2002, 2004a) further noticed that the southward propagating wave could be reflected when encountering the sharply deepened bottom topography south of the strait and suggested that the reflected wave is the main component of the northward propagating wave and that the contribution of the SCS is negligible. Yu et al. (2015) completed an extensive numerical study of the formation of the $M_2$ tide in the strait with a special focus on the asymmetric nature in the cross-strait direction.

The existing studies almost all employed data analysis and numerical modelling, except that some simple dynamical analyses were performed using one-dimensional solutions to explain the model results by Jan et al. (2002) and Yu et al. (2015). The purpose of the present study is to establish two-dimensional analytical models using an extended Taylor method (see Section 2 for details). In the analytical models, the classical Kelvin waves and Poincaré modes in idealized basins are used to approximately represent the tides in the natural basin. This enables us to estimate the strengths of the southward and the northward waves to reveal the role of each classical wave in the formation of the tides in the strait and to clarify how the

waves are generated. In particular, we can roughly estimate the relative importance of the reflected wave at steep topography versus the incident wave from the LS in the formation of the northward Kelvin wave in the TS.

The Taylor problem is a classical tidal dynamic problem (Hendershott and Speranza, 1971). Since his pioneering work, Taylor's method has been subsequently developed and applied to many sea areas (e.g., Table 1 of Roos et al., 2011). In the previous applications, most of the studied basins have a closed end that can almost perfectly reflect the incident tidal wave, thus closely retaining the phase of the tidal elevation. In contrast, the topographic step south of the TS acts as a permeable interface that can only partially reflect the incident wave, and furthermore, the elevation phase of the reflected wave is changed by nearly 180° at the step (see Section 5.5 of Dean and Dalrymple, 1984). Therefore, the strait is also a locality of particular interest for the application of Taylor's method.

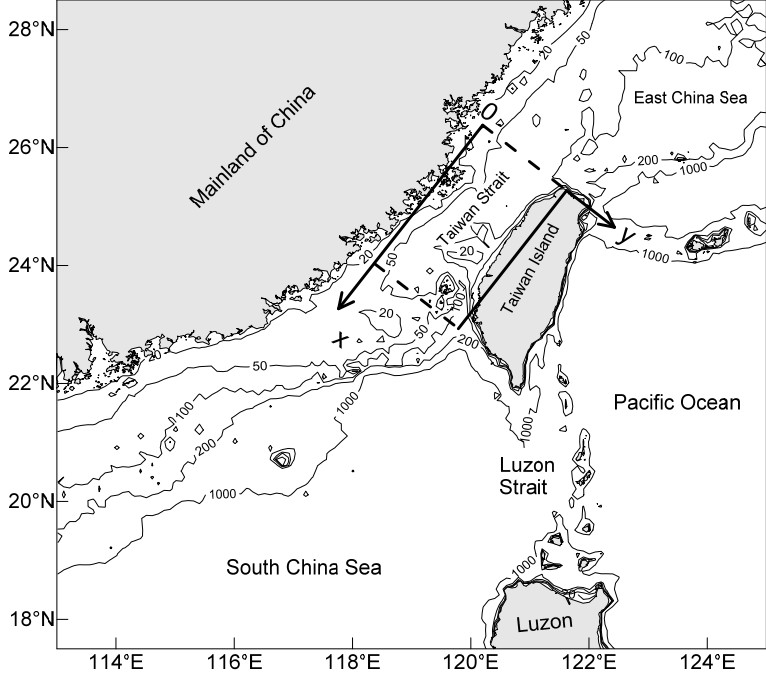

**Figure 1.** Bathymetric chart of the Taiwan Strait and its neighbouring area; the rectangle indicates the idealized model basin representing the Taiwan Strait. Isobaths are in metres (based on ETOPO1 from the US National Geophysical Center).

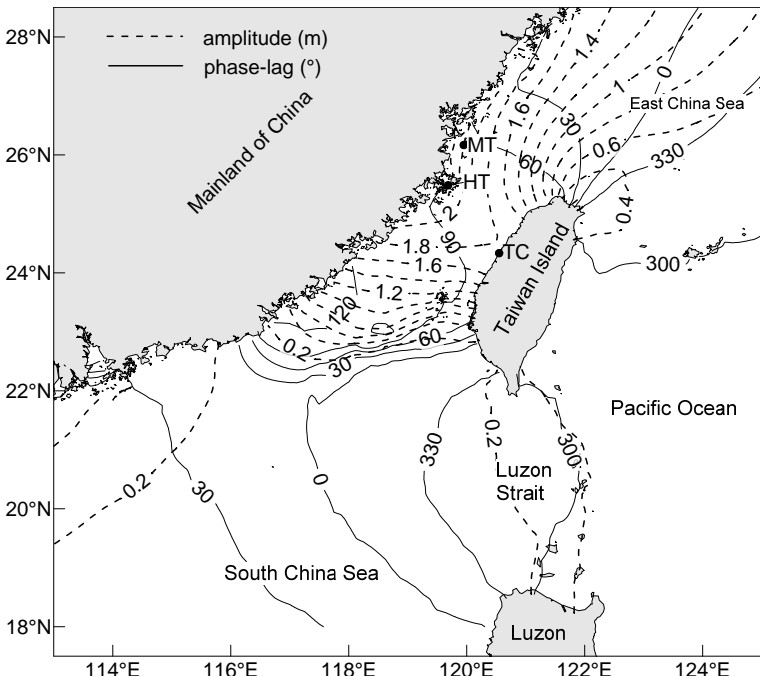

**Figure 2.** The $M_2$ tidal system in the Taiwan Strait and its neighbouring area (based on DTU10, see Cheng and Andersen, 2011). Solid lines represent the Greenwich phase-lag (in degrees), and dashed lines represent amplitude (in metres). (MT-Matsu, HT-Haitan, TC-Taichung)

## 2 Model formulation and solution method

Taylor (1922) first presented an analytical solution for tides in a semi-infinite rotating rectangular channel of uniform depth to explain the existence of amphidromic systems in gulfs. His solution showed that the tide in such a channel can be represented by the superposition of an incident Kelvin wave, a reflected Kelvin wave and a family of Poincaré modes trapped near the closed end. In 1925, Defant simplified Taylor's solution approach by applying the collocation method (see Defant, 1961, pp. 213-215). In the original version of Taylor's problem, as well as the Defant's approach, the friction and open boundary condition were left out of consideration. Fang and Wang (1966) and Rienecker and Teubner (1980) extended the Taylor problem by taking friction into consideration in the governing equations. The introduction of friction can explain why the amphidromic point in the northern hemisphere shifts from the central axis toward the right, as seen from the closed end and looking seawards. The mechanism of the shift of the amphidromic point was also explained by Hendershott and Speranza (1971), in which the dissipation was assumed to occur at the closed end of the basin rather than during the wave propagation. Fang et al. (1991) further extended the Taylor problem by introducing the open boundary condition, enabling solutions accounting for the finite length of the basin. Jung et al. (2005), Roos and Schuttelaars (2011), and Roos et al. (2011) further extended the Taylor method to model tides in multiple rectangular basins. The solution method used in the present study is basically the same as Fang et al. (1991), but with minor correction and generalization, as was done in studies of Jung et al. (2005), Roos and Schuttelaars (2011), and Roos et al. (2011). The analytical method initiated by Taylor and developed afterward is called an extended Taylor method in this paper.

## 2.1 Governing equations and boundary conditions

The governing equations used in this study are as follows:

$$\begin{cases} \frac{\partial \tilde{u}}{\partial t} - f\tilde{v} = -g\frac{\partial \tilde{\zeta}}{\partial x} - \gamma \tilde{u} \\ \frac{\partial \tilde{v}}{\partial t} + f\tilde{u} = -g\frac{\partial \tilde{\zeta}}{\partial y} - \gamma \tilde{v} \\ \frac{\partial \tilde{\zeta}}{\partial t} = -h\left[\frac{\partial \tilde{u}}{\partial x} + \frac{\partial \tilde{v}}{\partial y}\right] \end{cases} \tag{1}$$

where $t$ represents time; $(x, y)$ are the Cartesian coordinates; $(\tilde{u}, \tilde{v})$ are the depth-averaged velocity components in the

$(x, y)$ directions; $\tilde{\zeta}$ is the tidal elevation; $h$ is the water depth, assumed uniform; $\gamma$ is the frictional coefficient, taken as a

constant; $g = 9.8 \text{ ms}^{-2}$ is the acceleration due to gravity; and $f$ is the Coriolis parameter, also taken as a constant due to

the smallness of the study area. The equations in (1) are two-dimensional linearized shallow water equations on an $f$-plane

with the momentum advection neglected. The equations are the same as those used in the work of Taylor (1922), except that

the bottom friction is incorporated, as in Fang and Wang (1966) and Rienecker and Teubner (1980). When a monochromatic

wave is considered, $(\tilde{\zeta}, \tilde{u}, \tilde{v})$ can be expressed as follows:

$$(\tilde{\zeta}, \tilde{u}, \tilde{v}) = \text{Re}\{(\zeta, u, v)e^{i\sigma t}\} \tag{2}$$

where $(\zeta, u, v)$ are complex amplitudes of $(\tilde{\zeta}, \tilde{u}, \tilde{v})$, respectively, and $\sigma$ is the angular frequency of the wave, $i \equiv \sqrt{-1}$.

For this wave, the equations in (1) reduce as follows:

$$\begin{cases} (\mu + i)u - vv = -\frac{g}{\sigma}\frac{\partial \zeta}{\partial x} \\ (\mu + i)v + vu = -\frac{g}{\sigma}\frac{\partial \zeta}{\partial y} \\ \zeta = \frac{ih}{\sigma}\left[\frac{\partial u}{\partial x} + \frac{\partial v}{\partial y}\right] \end{cases} \tag{3}$$

in which $\mu = \frac{\gamma}{\sigma}$, $v = \frac{f}{\sigma}$.

Considering a rectangular basin with two parallel sidewalls of length $L$ and with a width $B$, we placed the $x$ axis along

a sidewall and the $y$ axis perpendicular to the $x$ axis and pointing to the other sidewall. Thus, the basin is confined by $x = 0, L$ and $y = 0, B$, respectively. The boundary conditions along the sidewalls are taken as follows:

$$v = 0 \text{ at } y = 0 \text{ and } y = B \tag{4}$$

within $x \in (0, L)$. Along the cross sections, $x = 0$ and $x = L$, various choices of boundary conditions are applicable

depending on the problem concerned:

$$u = 0, \text{ if the cross section is a closed boundary;} \tag{5}$$

$$u = \pm\sqrt{\frac{g}{(1-i\mu)h}}\zeta, \text{ if the free radiation in the positive/negative } x \text{ direction occurs on the cross section;} \tag{6}$$

$$\zeta = \hat{\zeta}, \text{ if the tidal elevation is specified as } \hat{\zeta} \text{ along the cross section;} \tag{7}$$

and/or

$$\zeta_A = \zeta_B \text{ and } u_A h_A = u_B h_B, \text{ if the cross section is a connecting boundary of two basins } A \text{ and } B, \text{ each with a different}$$

uniform depth of $h_A$ and $h_B$. $\tag{8}$

The equations in (8) show the matching conditions accounting for sea level continuity and volume transport continuity, respectively. The individual equations (5) to (8), or their combination, may be used as boundary conditions for the cross sections. The relationship between $u$ and $\zeta$ shown in Eq. (6) is based on the solution for progressive Kelvin waves in presence of friction, which will be given in Eqs. (9) and (10) below. If $\mu \ll 1$, a simpler equation $u = \pm\sqrt{\frac{g}{h}}\,\zeta$ can be used to replace Eq. (6).

## 2.2 General solution and collocation method

The governing equations in (3) have only the following four forms satisfying the sidewall boundary condition of (4) (see Fang et al., 1991, an error in the equation for $\beta$ in their paper has been corrected here. Note: the error occurred during their preparation of the manuscript and the correct expression was used in their computations):

$$\begin{cases} v_1 = 0 \\ u_1 = -a\ \exp(\alpha y + i\beta x) \\ \zeta_1 = \frac{\beta}{\sigma} ha \exp(\alpha y + i\beta x) \end{cases} \tag{9}$$

$$\begin{cases} v_2 = 0 \\ u_2 = b\ \exp[-(\alpha y + i\beta x)] \\ \zeta_2 = \frac{\beta}{\sigma} hb\ \exp[-(\alpha y + i\beta x)] \end{cases} \tag{10}$$

$$\begin{cases} v_3 = \sum_{n=1}^{\infty} \kappa_n \sin r_n y \exp(-s_n x) \\ u_3 = \sum_{n=1}^{\infty} \kappa_n (A_n \cos r_n y + B_n \sin r_n y) \exp(-s_n x) \\ \zeta_3 = \frac{ih}{\sigma} \sum_{n=1}^{\infty} \kappa_n \left( C_n \cos r_n y + D_{1,n} \sin r_n y \right) \exp(-s_n x) \end{cases} \tag{11}$$

and

$$\begin{cases} v_4 = \sum_{n=1}^{\infty} \lambda_n \sin r_n y \exp[-s_n(L-x)] \\ u_4 = \sum_{n=1}^{\infty} \lambda_n (A_n' \cos r_n y + B_n' \sin r_n y) \exp[-s_n(L-x)] \\ \zeta_4 = \frac{ih}{\sigma} \sum_{n=1}^{\infty} \lambda_n (C_n' \cos r_n y + D_n' \sin r_n y) \exp[-s_n(L-x)] \end{cases} \tag{12}$$

where

$$\alpha = \frac{\nu}{(1-i\mu)^{\frac{1}{2}}} k \tag{13}$$

$$\beta = (1 - i\mu\ )^{\frac{1}{2}} k \tag{14}$$

$$r_n = \frac{n\pi}{B} \tag{15}$$

$$s_n = (r_n^2 + \alpha^2 - \beta^2)^{\frac{1}{2}} \tag{16}$$

in which $k = \sigma/c$ is the wave number, with $c = \sqrt{gh}$ being the wave speed of the Kelvin wave in the absence of friction. In Eq. (16), $s_n$ has two complex values for each $n$, and here, we choose the one that has a positive real part. To satisfy equations in (3), $(A_n, B_n, C_n, D_n)$ and $(A_n', B_n', C_n', D_n')$ should be as follows:

$$A_n = \frac{[(\mu+i)^2 + \nu^2] r_n s_n}{(\mu+i)^2 r_n^2 + \nu^2 s_n^2} \tag{17}$$

$$B_n = \frac{\nu(\mu+i)(\alpha^2 - \beta^2)}{(\mu+i)^2 r_n^2 + \nu^2 s_n^2} \tag{18}$$

$$C_n = r_n - s_n A_n \tag{19}$$

$$D_n = -s_n B_n \tag{20}$$

$$A_n' = -A_n \tag{21}$$

$$B_n' = B_n \tag{22}$$

$$C_n' = C_n \tag{23}$$

$$D_n' = -D_n \tag{24}$$

Eqs. (9) and (10) represent Kelvin waves propagating in the $-x$ and $x$ directions, respectively; Eqs. (11) and (12)

represent two families of Poincaré modes trapped at the cross sections $x = 0$, $L$, respectively. Coefficients $(a, b, \kappa_n, \lambda_n)$ are

related to amplitudes and phases of Kelvin waves and Poincaré modes. These coefficients are to be determined by boundary

conditions.

The collocation method is convenient when determining the coefficients $(a, b, \kappa_n, \lambda_n)$. The calculation procedure can be

as follows. First, we truncate the family of Poincaré modes, Eqs. (11) and (12), at the N-th order, so that the number of

undetermined coefficients for Poincaré modes is $2N$ and the total number of undetermined coefficients (plus those for

Kelvin waves) is thus $2N + 2$. To determine these unknowns, we take equally spaced $N + 1$ dots, called collocation points,

located at $y = \frac{B}{2(N+1)}, \frac{3B}{2(N+1)}, \ldots, \frac{(2N+1)B}{2(N+1)}$ on both the cross sections $x = 0$ and $L$. At these points, one of the boundary

conditions given by Eqs. (5) - (8) should be satisfied. This yields $2N + 2$ equations. By solving this system of equations,

we can obtain $2N + 2$ coefficients $(a, b, \kappa_n, \lambda_n)$. Since the high-order Poincaré modes decay from the boundary very

quickly as can be seen from large $s_n$ in (11) and (12), it is generally necessary to retain only a few lower order terms.

## 3 Application to the Taiwan Strait

### 3.1 Model configuration and solution

In this section, we will first establish an idealized analytical model for the TS. The strait is idealized as a rectangular basin

with two sidewalls roughly along the China mainland and Taiwan coastlines, as shown in Fig. 1. The width and length of the

model domain are taken as $B$=200 km and $L$=330 km, respectively. The depth is taken as $h = 52$ m, a mean depth

calculated based on ETOPO1. We place the origin of the coordinates at the northernmost corner of the rectangle, the $x$ axis

along the mainland coast, and the $y$ axis in an offshore direction. The axis of the strait is toward the south to southwest.

However, to keep it short, we will hereafter simply use "south" to refer "south to southwest", and similarly for other

directions. The Coriolis parameter $f$ is taken as $0.594 \times 10^{-4} \text{s}^{-1}$, corresponding to a latitude of $\varphi = 24°N$. The angular

frequency of the M$_2$ tide is $1.4052 \times 10^{-4} \text{s}^{-1}$. The friction coefficient $\gamma$ can be estimated from the relation $\gamma = C_D \left(\frac{8}{3\pi}\right) \frac{U}{h}$,

in which $C_D$ and $U$ represent the drag coefficient and amplitude of the M$_2$ tidal current, respectively (e.g., Chapter 8 of

Dronkers, 1964). In this study, we take $C_D$ =0.0026 and $U$ =0.5 m/s based on the numerical results of Fang et al. (1984),

and then, $\mu = \gamma/\sigma$ is approximately equal to 0.15. From these parameter values, we can obtain the wavelength of the M$_2$

Kelvin wave as 1009 km. Since the basin width is smaller than half of the Kelvin wavelength, the Poincaré modes can only exist in a bound form (Godin, 1965; Fang and Wang, 1966). The e-folding length of decay of the lowest Poincaré mode is approximately 63 km, that is, the amplitude of this mode reduces to approximately 37% relative to its maximum value at a distance of 63 km away from the boundary. Equivalently, it may also reduce to approximately 20% relative to its maximum value at a distance of 100 km. The length scales of decay for higher order Poincaré modes are even shorter.

In this study, the families of Poincaré modes are truncated at $N = 19$ and 20 collocation points set along both the northern and southern open boundaries. The boundary condition (7) is employed with the values of $\hat{\zeta}$ equal to the observed harmonic constants from the global tide model DTU10 (Cheng and Anderson, 2011).

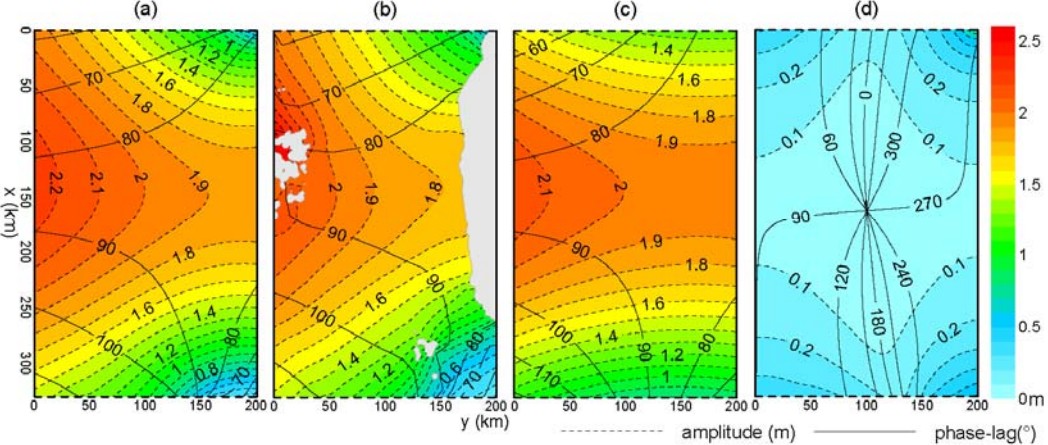

**Figure 3.** Tidal system charts for the $M_2$ constituent: (a) Present analytical model; (b) Observed distribution based on DTU10; (c) Contribution of Kelvin waves; (d) Contribution of Poincaré modes. Solid lines represent Greenwich phase-lag (in degrees); dashed lines represent amplitude (in metres).

The obtained analytical solution of the $M_2$ constituents is shown in Fig. 3a. For comparison, the observed $M_2$ tidal system chart based on DTU10 is also shown in Fig. 3b. Although the complicated bottom topography and the irregular coastlines are greatly simplified, the analytical model still agrees well with the observation. The observed tidal regime features significantly greater amplitudes along the mainland coast than along the Taiwan coast, showing the cross-strait asymmetry. The phase-lags near the mainland coast increase from north to south, showing a progressive wave nature, while those near the middle Taiwan coast have only small changes, showing a standing wave nature. That is, along the Taiwan coast the wave propagates southward in the northern area and propagates northward in the southern area. The largest amplitudes are roughly along the cross section of 150 km and appears as an anti-nodal band. The phase-lags in this band range from 80° to 90°. These features have all been reproduced in the analytical model.

### 3.2 Kelvin waves and Poincaré modes

To reveal the relative importance of the Kelvin waves and Poincaré modes in the model, the superposition of two Kelvin waves is given in Fig. 3c and that of the Poincaré modes is given in Fig. 3d. The contribution of the Poincaré modes is observed to be much smaller than that of the Kelvin waves. The tidal system chart constructed using only superposed Kelvin

waves (Fig. 3c) resembles the complete model (Fig. 3a) and the observation (Fig. 3b) quite well, though the inclusion of the Poincaré modes improves the model to a certain degree. From Fig. 3a we can see that the difference between the highest amplitude on the west sidewall and that on the east sidewall in the anti-nodal band is approximately 0.4 m, while the corresponding difference shown in Fig. 3c is approximately 0.2 m. Thus, approximately half of the cross-strait asymmetry is explained by the superposition of two oppositely propagating Kelvin waves, with the southward one being stronger than the one moving northward. Here, both the Coriolis force and the weaker northward wave are the major factors. The superposition of Poincaré modes in this band has an amplitude of approximately 0.1 m on both sides and has nearly the same phase-lag as the superposed Kelvin wave on the west and a nearly opposite phase-lag to the superposed Kelvin wave on the east. Therefore, the superposed Poincaré modes play a role to increase the amplitudes in the west and reduce the amplitudes in the east and hence enhances the asymmetry. The superposed Poincaré modes make nearly the same contribution to the cross-strait asymmetry as the superposed Kelvin wave.

From the comparison, we find that the amplitude variation along the northern boundary in Fig. 3c is less than that in Fig. 3a. This shows that near the boundary, the Poincaré modes are of a certain importance. The existence of the Poincaré modes is related to the fact that the $M_2$ tide is from the Pacific Ocean; its amplitude increases from the deeper outer shelf toward the shallower inner shelf. This amplitude variation cannot be completely represented by the superposed Kelvin wave at a uniform depth, and a superposed Poincaré modes are necessary to compensate for their difference. The situation at the southern boundary is similar. The distribution of the superposed Poincaré modes in the anti-nodal band is clearly related to those at the northern and southern openings (Fig. 3d). Yu et al. (2015) suggested that the orientation of the topographic step south of the strait was not perpendicular to the strait axis but has an angle. This might cause the reflected wave to propagate toward the mainland coast and thus amplify the tides there. The present solution indicates that the obliqueness of the topographic step south of the TS may also play a role in the formation of the cross-strait asymmetry, as suggested by Yu et al. (2015), but it seems not to be a controlling factor.

The obtained analytical solution enables us to see the magnitudes and characteristics of both the southward and northward Kelvin waves. These two oppositely propagating waves, which correspond to Eqs. (9) and (10) respectively, are displayed separately in Figs. 4a and 4b. From Fig. 4a, we see that the phase-lag of the southward wave increases from north to south. The amplitude deceases from north to south due to friction and from west to east due to the Coriolis effect. The characteristics of the northward wave are the opposite. The area mean amplitude of the southward wave is 1.18 m, while that of the northward wave is 0.84 m, smaller than the former by 0.34 m. Along the western sidewall, the amplitudes of the southward wave range from approximately 1.4 m to 1.6 m, while those of the northward wave range from approximately 0.6 to 0.7 m; thus, the superposition of the waves is dominated by the former and appears as a southward progressive wave. Around the cross section $x \approx 150$ km, the phase-lags of the southward and northward waves are nearly equal, between 80° and 90°. Thus, the superposed tides here have the greatest amplitudes equal to the sum of the amplitudes of these two waves,

exceeding 2.1 m, as already seen in Fig. 3c. Along the eastern sidewall, however, the differences in amplitudes of the southward and northward waves are much smaller, and thus, the superposition of the waves tends to appear as a standing wave. Around the point $x \approx 150$ km, the phase-lags of the southward and northward waves are also nearly equal. Thus, the amplitude of the combined tide is also relatively large, equal to the sum of the amplitudes of these two waves, but now it is only slightly greater than 1.9 m, which is smaller than the corresponding value at the western sidewall.

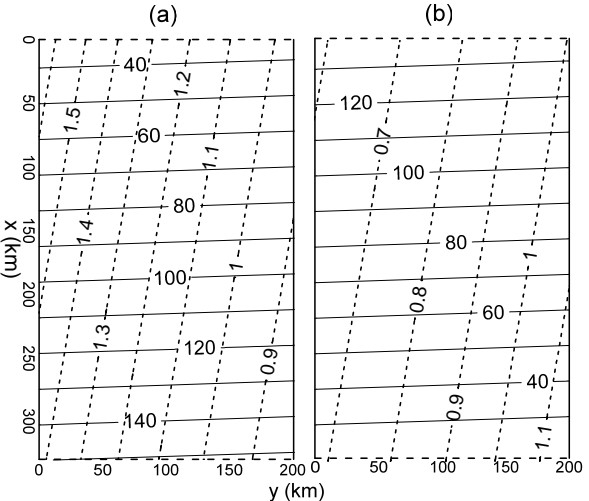

**Figure 4.** Southward (a) and northward (b) propagating Kelvin waves**.** Solid lines represent Greenwich phase-lag; dashed lines represent amplitude (in metres).

**4 Formation mechanism of the northward Kelvin wave in the Taiwan Strait**

In the preceding section, we have shown that the northward Kelvin wave is weaker than the southward wave on average, but they have a similar magnitude along the Taiwan coast. In this section, we will examine the formation mechanism of the northward Kelvin wave. There are two possible origins for the northward Kelvin wave in the TS. One is the reflection of the southward wave at sharply deepened topography and another is an incident wave from the Luzon Strait propagating toward the TS. In the following, we examine their respective contributions by using the extended Taylor models.

**4.1 Reflection of the incident wave from the East China Sea at the topographic step**

Three experiments have been carried out to explore the formation mechanism of the northward Kelvin wave in the TS. The first experiment (denoted as Ex. 1) has the model geometry shown in Fig. 5a. The TS is represented by area A, with the width and depth equal to the above single area model. Since the topographic step is located away from the southern boundary of the single area model domain (Fig. 1), we extend the length of the area to 400 km. Area B represents the deep basin south of the topographic step, and the water depth of the deep basin is taken as 1000 m, as was done in Jan et al. (2002 and 2004). The purpose of this experiment is to examine the effect of the topographic step in reflecting the incident wave from the ECS. The experimental design for area A is similar to that of Roos and Schuttelaars (2011): a southward Kelvin wave is specified to be identical to the single basin solution, as shown in Fig. 4a in the preceding section. The Poincaré modes trapped at the cross

section $x = 0$ are neglected, while those trapped at the cross section $x = 400$ km are retained. The matching condition (8) is applied at the connecting boundary of areas A and B, and the radiative condition (6) is used at the southernmost opening.

Figure 5b displays the solution of Ex.1. It can be seen that the basic pattern of the tidal regime is similar to that of the single area model solution shown in Fig. 3c. In particular, there is again an anti-nodal band near $x = 150$ km, though the amplitudes in this band produced by this experiment are smaller than those given in Fig. 3c. The smallest amplitudes appear along the connecting cross section, showing that a nodal band exists there. Therefore, the anti-node is located approximately 250 km away from the topographic step. The wavelength of the $M_2$ tide in a channel of a uniform depth of 52 m is equal to 1009 km, and so, the distance between the anti-node and the topographic step is equal to one quarter of the wavelength. This result further implies that if the channel were 500 km long, resonance would occur. However, Taiwan Island is approximately 380 km long and is not able to support a resonance for the $M_2$ constituent. In fact, the resonant period of the TS is 13.5 h, according to the experiments performed by Cui et al. (2015), which is almost the same as one of the resonant periods of the ECS (13.7 h, obtained by Cui et al., 2015). This means that the tidal response in the TS is not independent, but rather closely related to the tides in the ECS.

The southward and northward Kelvin waves obtained from Ex. 1 are shown in Figs. 5c and 5d, respectively. Comparison of these figures with Figs. 4a and 4b indicates that in area A, the southward wave is identical, but the northward wave from Ex. 1 is weaker. For the area $x = 0$ to 330 km and $y = 0$ to 200 km, the area mean amplitude of the northward Kelvin wave is 0.57 m, which is smaller than the single area model value by 32%. In area B, the amplitudes of the transmitted southward Kelvin wave are approximately 0.4 m, and those of the northward wave are negligible. An important difference in the co-phase-lag distributions is that Figs. 3a-c show a northward propagation along the southern part of the eastern sidewall, while Fig. 5b does not have such a feature. This is because in the single area case, the amplitudes of the northward Kelvin wave are greater than those of the southward Kelvin wave in this area (Figs. 4a, 4b), while in Ex. 1, this situation does not occur (Figs. 5c, 5d).

The relative magnitudes of the incident and the reflected and transmitted Kelvin waves can be evaluated by comparing their amplitudes along the connecting cross section at $x = 400$ km. The sectional mean amplitudes for the incident, reflected and transmitted waves, $H_i$, $H_r$ and $H_t$ are 1.06, 0.64 and 0.40 m, respectively (Figs. 5c, 5d). Thus, the ratios $H_r / H_i$ and $H_t / H_i$ are equal to 0.61 and 0.37 respectively. The corresponding values based on the theory ignoring the earth's rotation can be calculated from $\frac{H_r}{H_i} = \frac{1-\rho}{1+\rho}$ and $\frac{H_t}{H_i} = \frac{2\rho}{1+\rho}$ with $\rho = \sqrt{h_A / h_B}$ (e.g., Dean and Dalrymple, 1984, p. 144). Substitution of the present model depths into these equations yields $H_r / H_i = 0.63$ and $H_t / H_i = 0.37$. This indicates that the magnitude of the reflected waves in the two-dimensional case with the earth's rotation being taken into account is smaller than that based on the theory with the earth's rotation being ignored.

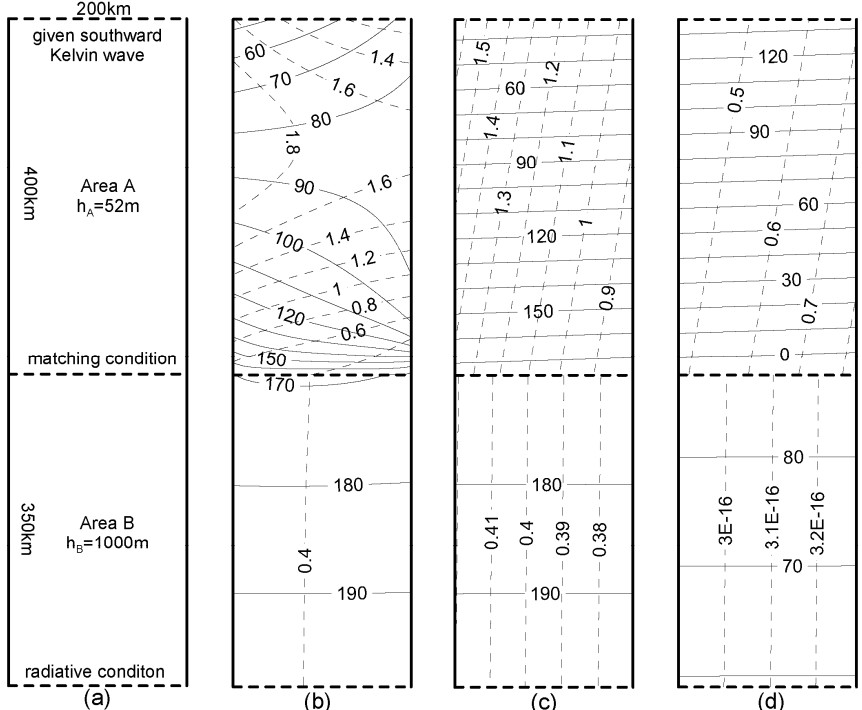

**Figure 5.** Model domain and boundary conditions of Ex. 1 (a); solution of Ex. 1 (b); southward Kelvin waves (c); northward Kelvin waves (d). Solid lines represent Greenwich phase-lag (in degrees); dashed lines represent amplitude (in metres).

## 4.2 Influence of the shelf region southwest of the Taiwan Strait

5    From Fig. 1, we can see that there is a narrow shelf along the mainland coast. To simulate the effect of the narrow shelf on the

tides in the TS, we performed a second experiment, numbered Ex. 2. In this experiment, the deep basin has moved 60 km

eastward, allowing the tides in the shallow basin to freely radiate southward as shown in Fig. 6a. The radiative condition (6)

is retained along the southernmost opening. The results of Ex. 2 are given in Fig. 6. It can be seen that the tides in area A have

only small changes, though the deep basin has moved 60 km eastward. Observable changes can only be found in area B where

10   the tidal amplitudes are slightly reduced.

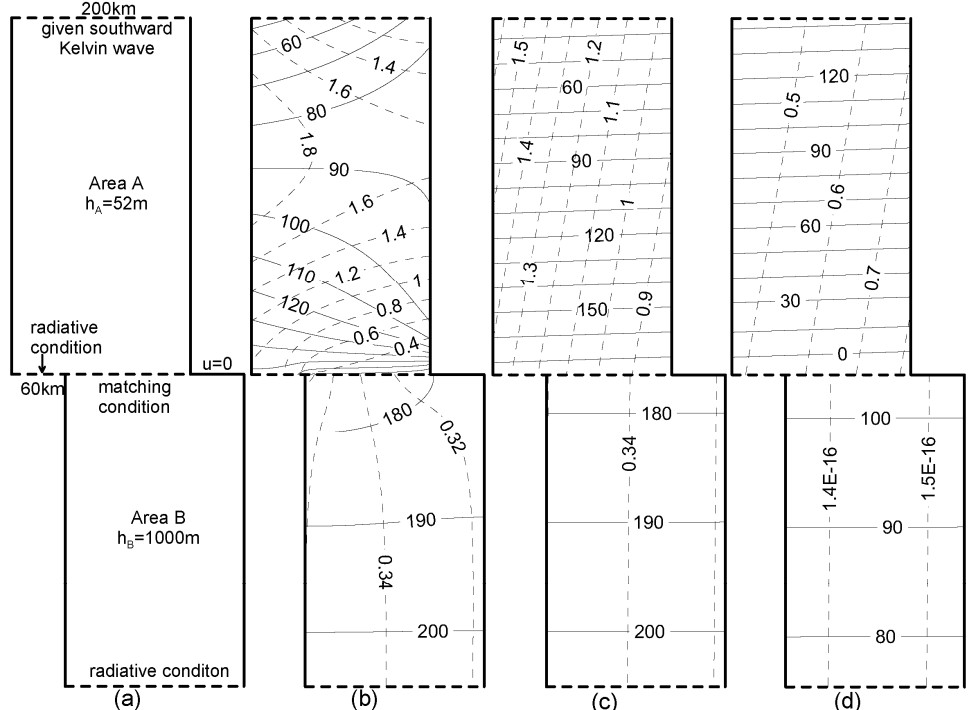

**Figure 6.** Same as Figure 5, but for Ex. 2.

## 4.3 Influence of the Luzon Strait forcing

The purpose of performing a third experiment, numbered Ex. 3, is to consider the tidal input from the LS. The major difficultly

in including the LS input in Taylor's model for the TS is that the LS has a meridional orientation, while Taylor's model does

not allow any part of the sidewalls to open. Here, we will use a rather crude model to solve this issue. We use the same model

domain as Ex. 2, but the radiative boundary condition (6) is retained only for the west segment of the southernmost opening,

and the boundary condition (7) is applied to the remaining east segment of the opening. From Fig. 1, we can see that the cross

section from the mainland shelf to the LS is much longer than the width of the LS. Thus, in our model, we take the lengths of

the west and east segments to be 120 km and 80 km, respectively, as shown in Fig. 7a. In addition, from Fig. 2, we observe

that the tidal amplitude along the LS is roughly 0.2 m, and the phase-lag is approximately 310°. Since a significant portion of

the incident wave from the LS propagates toward the SCS deep basin (e.g., Fang et al., 1999; Yu. et al., 2015), we use a 0.1 m

amplitude and 310° phase-lag as an open boundary condition for the east segment of the southernmost opening in Ex. 3. The

model results are given in Figs. 7b to 7d. From Fig. 7b, we can see that the amplitudes of the tide in area A now become greater

than the results of Ex. 2 (Fig. 6b), and a northward propagating character can be seen in the south-eastern portion of area A.

These improvements can be attributed to the increased amplitudes of the northward Kelvin wave (Figs. 6d, 7d).

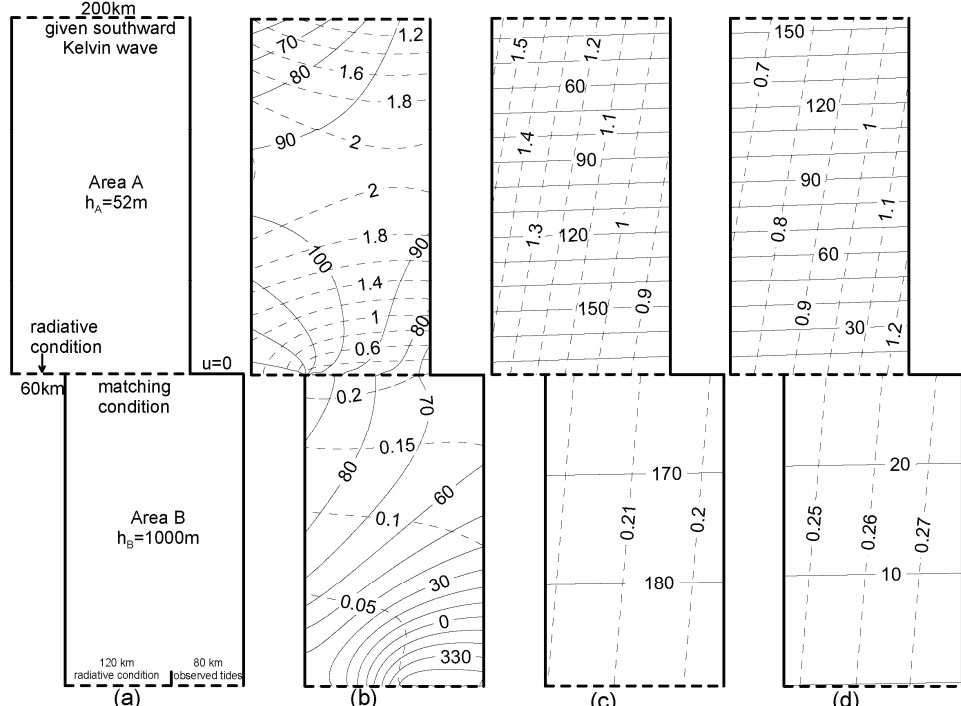

**Figure 7.** Same as Figure 5, but for Ex. 3.

## 5 Summary and discussion

In the present study, we first established an analytical model for the $M_2$ tide in the TS using the extended Taylor method. The

superiority of the analytical solution is that the tides can be decomposed into a southward Kelvin wave, a northward Kelvin

wave, and two families of Poincaré modes, providing a deeper insight into the dynamics of the tides in the area. Though the

coastlines and bottom topography are greatly simplified, the model-produced pattern resembles the observed tidal regime

quite well. We then carried out several experiments to examine the formation mechanism of the northward propagating

wave, especially the roles of the abruptly deepened bottom topography south of the TS and the tidal forcing in the LS in the

formation of the northward wave. From this study, we have obtained the following results.

The $M_2$ tide in the TS can be basically represented by the superposition of a southward propagating and a northward

propagating Kelvin wave, with the former being stronger than the latter. The superposed Kelvin waves give an anti-nodal

band near the cross-strait transection, roughly from Haitan Island to Taichung. The maximum amplitude on the mainland

side is greater than that on the Taiwan side, showing the cross-strait asymmetry. Therefore, the observed features can be

reproduced by the superposition of a stronger southward propagating and a weaker northward propagating Kelvin wave. In

this regard, the Coriolis force and the weaker northward wave play essential roles.

Inclusion of the Poincaré modes in the analytical model improves the model results: the east to west increase in

amplitudes along the northern and southern openings is better reproduced; and in particular, the Poincaré modes make

approximately the same contribution as the Kelvin waves to the cross-strait asymmetry in the anti-nodal band.

The reflection of the southward wave at the abruptly deepened topography south of the TS is a major contribution to the formation of the northward propagating wave in the strait. However, the reflected wave is slightly weaker than that obtained from the analytical solution with open boundary conditions determined by the observations. Inclusion of the tidal forcing at the LS strengthens the northward Kelvin wave in the TS and thus improves the model result. This indicates that the LS forcing is of some (but lesser) importance to the $M_2$ tide in the TS.

The analytical solutions can help us to understand the dynamics of tidal motion in the TS, but there are some limitations. For example, the LS is located on the east side of the study area, while the Taylor model does not allow for a forcing on the sidewalls, and thus, we are bound to let a part of southern opening represent the LS (Fig. 7a). In addition, we have assumed that the water depth changes from 52 m to 1000 m immediately at the connecting cross section without considering the existence of the continental slope at that location. The obliqueness of the orientation of the topography step relative to the cross-strait direction is also ignored. These approximations will induce uncertainty in the results for the magnitude of the reflected wave.

*Acknowledgements.* This study was supported by the NSFC-Shandong Joint Fund for Marine Science Research Centers (Grant No. U1406404), the National Natural Science Foundation of China (Grant No. 41706031), the Basic Scientific Fund for National Public Research Institutes of China (Grant No. 2014G15), and the National Key Research and Development Program of China (Grant No. 2017YFC1404201). The authors sincerely thank Dr. Huthnance and two anonymous Referees for their constructive comments and suggestions, which are of great help in improving our study.

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
