# Peer review of "An analytical study of M2 tidal waves in the Taiwan Strait using an extended Taylor method"

_Ocean Science, 2017_

## Referee Comment (RC1) · Anonymous Referee #1 · 28 Aug 2017

Dear Editor,

The authors present an interesting study into the tidal dynamics of the Taiwan Strait. Particularly, they apply a so-termed 'extended' version of the classical 'Taylor method' to reproduce and explain the amphidromic pattern of the semi-diurnal tide in that region. The word 'extended' here refers to the treatment of the (open) boundaries and the inclusion of bottom friction. This leads to an analysis of two Kelvin waves, propagating southward and northward, the superposition of which largely determines the amphidromic pattern in the Taiwan Strait. As to the sources that may contribute to the northward Kelvin wave, the authors conduct a further analysis in which the model is extended in various ways. To be honest, I find this part a bit far-fetched, simply because

the rather 'crude' (as the authors acknowledge themselves) geometrical choices made here clearly ignore the true geometry of the sea surrounding the Taiwan Strait, particularly regarding coastlines. This makes the conclusions of this part less convincing to me, which is actually my first concern of this study. The same applies in my opinion to statements about the "superiority" of this approach in the conclusions. Other aspects that – in my opinion – require clarification or improvement deal with (1) description of the study site, (2) literature review, (3) model formulation, (4) comparison with observations, (5) interpretation of Kelvin and Poincaré modes, and (6) phrasing. These points are detailed below. Overall, I think the topic of the paper is appropriate for OSD. The novelty of the work is apparent, but the my concerns on how this has been done are substantial. Therefore, my overall recommendation is major revision.

Anonymous, 28 August 2017.

**1) Description of the study site** may be extended by presenting the relative importance of other tidal constituents (S2, K1, O1), e.g. expressed in the value of the form factor $F$. Why did you consider the M2-tide only? And what is known about the (magnitude of the) tidal currents? This helps interpretation compared to other tidal basins around the world.

**2) Literature review** should in my opinion be improved in certain respects.

- The large number of references on tides in the Taiwan Strait makes me wonder what has been found in those studies. . .

- Page 2, Line 12: "was the main component" –> "is the main component".

- Upon first introduction in Line 18, The extended Taylor method (when using "the", please remove the "'s") requires a reference and an explanation of what 'extended' means here.

- Roos & Velema should in fact be Roos et al (there are more co-authors). Also,

unlike suggested by the authors here, the presence of the Dover strait in the south is in fact an open boundary.

- I cannot find Table 1 in the .pdf-file that for this review.

- Hendershott & Speranza (Deep Sea Res 1971) is worthwhile mentioning as they followed a similar approach to study the Gulf of California (two Kelvin waves)

- Because of the depth-step, one may consider reference to Roos & Schuttelaars (Ocean Dyn 2011)

- Figure 2: "amphidromic chart" seems better, because it is both co-tidal and co-range information that is plotted here. Also: is it Chen and Andersen or Cheng and Andersen?

**3) Model formulation** contains some inaccuracies. First of all, the title of section 2 does not really cover the content. I think "Model formulation and solution method" is more appropriate.

- Please mention the important simplifications/approximations made here. This is a linear depth-averaged model, the validity of which is relevant. I think this should be discussed at some point.

- The pressure gradients in Eq.(1) should have spatial derivatives ($\partial/\partial x$ and $/\partial y$)

- Page 3, Line 8: "channel" –> "rectangular channel"

- Line 11: "by introducing a collocation method" –> "by applying a collocation method"

- Page 4, Line 5: "for open rectangular basins" –> "accounting for the finite length of the basin"

- Line 13: please mention "depth-averaged"

- Line 15: this approximation is known as the f-plane

- Line 16: "cosine wave" is perhaps better rephrased as "monochromatic"

- Line 17: please put brackets { and } after the real part: $\mathrm{Re}\left\{(\zeta, u, v)\exp(i\sigma t)\right\}$ and please introduce $(\zeta, u, v)$ as the complex amplitudes of the quantities introduced previously.

- Page 5, Line 4: please add "each with a different uniform depth $h_A$ and $h_B$"

- Line 9: would be nice if your radiation condition would include bottom friction. How large us mu typically?

- Line 23: the formula for wave speed also holds in absence of friction only... please reorder

- I would put the details of Eqs.(9)-(12) and Eqs. (17-24) in an appendix.

- Line 22: I think it is unnecessary to introduce $Q$, because you can immediately write $Q^2 = \beta^2 - -\alpha^2$.

**4) Comparison with observations** is purely visual, which raises some questions. First of all, how did you choose the basin dimensions, orientation? How do you actually project the true geometry, with curved coastlines, onto the rectangular model domain? And, as before: did you consider doing the same for other tidal constituents? Other than that, I find the title of Section 3 confusing, since the model has already been introduced in Section 2. I suggest to change the title into "Application to Taiwan Strait", because that is what is actually done in this section. Also, please avoid if statements when you specify coefficients (Page 7, line 2) an please replace "equal to" with an equality sign $=$.

**5) Interpretation of role of Kelvin and Poincare modes** can readily be deepened by further analysis. First of all, what is the wavelength of the Kelvin waves? (I see it is mentioned later on but already here it is relevant). And are the Poincare modes free or bound (from the depth and width values I guess they are all bound), and what is the typical length scale of decay of the lowest Poincare mode? This gives insight in the extent to which these modes affect the amphidromic pattern in the (interior of the) Taiwan Strait.

Also, I do not understand the statement that frictional force would be a major factor (as mentioned here and repeated in Section 5). I think this is not the case, in view of the mild amplitudes and large depths. Can you support this statement? I suspect you would still get a good fit if bottom friction were switched off.

- Page 7, Line 23: "inclusion of the Poincaré modes improves"

- Page 8, Line 10: Also possible is that the assumption of uniform depth is too restrictive in this Taylor approach...

- Page 8, Line 21: this is a basic statement about progressive waves and therefore not really insightful in my opinion.

- Page 10, I do not understand the statement on resonance. This may hold for closed basins, but here we have a topographic step...

**6) Phrasing** in general should be more precise in my opinion. For example, avoid the unnecessary and confusing use of the verb "can". My suggestion is to consult a native speaker of the English language with knowledge of the topic to revise the text. Here I explain what I mean by giving some suggestions to improve the abstract (line 8-22).

- Page 1, Line 8: "M2" –> "semidiurnal lunar (M2)"

- Line 8, "The extended Taylor's method", remove "'s": and is it sufficiently clear what this means?

- Line 10, "but" –> "and" (because this does not really signify a contradiction!)

- Line 10: "friction forces" –> "bottom friction"

- Line 16: "can further improve" is unclear. Better: "Inclusion of Poincaré modes further improves"

- Line 18: "can be reflected" –> "is reflected" (I guess this is what you mean)

- Line 21: same with "can" as in Line 18.

---

## Referee Comment (RC2) · Anonymous Referee #2 · 6 Sep 2017

General comments

This paper contains original contribution to analytical tide modeling using an Taylor' method. Although there are quite many thing to be clarified and improved, it is believed that authors can revise the manuscript without much difficulties. This paper is therefore recommended for the publication in OS with minor corrections.

Specific comments

Title

Pg.1, lines 1-2 Better to replace "the extended Taylor's method" to "an extended Taylor's method".

Abstract

Pg. 1, line 8: Again use "an extended Taylor's method Pg. 1, lines 21-22: The sentences are a little bit unnatural. Include how much the northward KW is strengthened, that is, quantitatively, saying it is of secondary importance.

1 Introduction

Pg. 2, line 4: The expression "anti-nodal band" is not familiar. Is there anybone to use the expression? Pg. 2, line 18: Again use "an extended Taylor's method Pg. 2, lines 27-28: What is the basis of "The statement "the topographic step south of the TS acts as a permeable interface which can only partially reflect the incident wave and .... by nearly 180 degree at the step". Previous studies? If not, authors already assume partial reflection of northward and southward waves at the step. More careful writing is needed. Pg. 4, line 9: Again use "an extended Taylor's method.

3. An analytical model for the Taiwan Strait

3.1 Model configuration and solution Pg. 7, lines 1-2: Reference for the friction coefficient formula is required. Pg. 7, line 5: The expression "observed harmonic constants from the global tide model" is a little bit strange. It is computed results not observed values. Recommend to change the expression. Pg. 7, line 6: In Figure 3(a) open boundary values appear to be somewhat different with Figure 2. Describe how the global model results were adopted to the analytical model. Pg. 7, lines 15-16: The statement "that is, the wave propagates southward in the northeast area and propagate northward in the southeast area" may be incorrect in strict sense. Figure 4 shows in whole area there are southward and northward Kelvin waves. ,Rewriting is needed. 3.2 Kelvin waves and Poincare modes Pg. 8, line 6: In Figure 3(a) open boundary values appear to be somewhat different with Figure 2. Describe how the global model results were adopted to the analytical model. Pg. 8, lines 9-10: The statement "the amplitude variation along the northern boundary ......owing to the fact that the M2 tide is from the Pacific Ocean" is obscure. The northern boundary values of Fig. 3c and

Fig.3a can be different from each other even though the M2 tide did not come from pacific Ocean. More careful discussions are required. Pg.8, lines 17-18: The statement "it is not a controlling factor" is too much definite. Obliqeness might be partly effective. Improve the statement.

4 Formation mechanism of the northward Kelvin wave in the Taiwan Strait

4.1 Reflection of the incident wave from the East China Sea at the topographic step Pg. 9, line 17: "observation taken from the global tide model" needs to be changed as mentioned earlier. Pg. 10. 4: Include reference for "the resonant period of the ECS (13.7h)".

5. Summary and discussion

Pg. 14, lines 1-2: Regarding the statement " the reflected wave is slightly weaker", authors implied that there is a partial reflection of southward KW at the step. It is noted that at the northern open boundary southward KW, northward KW and Poincare waves are all specified in Ex.1. If there is northward KW at the northern open boundary, there should be northward KW at the southern open boundary, regardless of topography. It is curious to know what will happen if only southward KW is imposed at the northern boundary in Ex.1. Additional experiment is recommended to clarify the partial reflection at the step. You may include results in appropriate place.

Technical corrections

Pg. 1, line 29: Better to replace "M2 amplitudes" to "M2 tide". Pg. 3, lines 5-6: Definition of phase lag needs to be added. Pg. 11, line 2: In Figure caption, there is no (d).

---

## Author Comment (AC1) · 30 Oct 2017

Dear Referee and Editor,

We sincerely thank the Referee for his careful reading of our manuscript, as well as the constructive comments and suggestions which are of great help for improving our study. We also sincerely thank the editor for the kind help with our paper. Every comment is considered and replied carefully in the response and the manuscript is revised accordingly. The derailed response and the revised manuscript are submitted as the PDF files in the supplement.

In our response, the following abbreviations are used:

OM - original manuscript; R1 – Revision #1 - an updated manuscript

[Figure]

Please also note the supplement to this comment:
https://www.ocean-sci-discuss.net/os-2017-61/os-2017-61-AC1-supplement.zip
* * *

---

## Author Response (AR1)

Dear Editor,

The authors present an interesting study into the tidal dynamics of the Taiwan Strait. Particularly, they apply a so-termed 'extended' version of the classical 'Taylor method' to reproduce and explain the amphidromic pattern of the semi-diurnal tide in that region. The word 'extended' here refers to the treatment of the (open) boundaries and the inclusion of bottom friction. This leads to an analysis of two Kelvin waves, propagating southward and northward, the superposition of which largely determines the amphidromic pattern in the Taiwan Strait. As to the sources that may contribute to the northward Kelvin wave, the authors conduct a further analysis in which the model is extended in various ways. To be honest, I find this part a bit far-fetched, simply because the rather 'crude' (as the authors acknowledge themselves) geometrical choices made here clearly ignore the true geometry of the sea surrounding the Taiwan Strait, particularly regarding coastlines. This makes the conclusions of this part less convincing to me, which is actually my first concern of this study. The same applies in my opinion to statements about the "superiority" of this approach in the conclusions. Other aspects that – in my opinion – require clarification or improvement deal with (1) description of the study site, (2) literature review, (3) model formulation, (4) comparison with observations, (5) interpretation of Kelvin and Poincaré modes, and (6) phrasing. These points are detailed below. Overall, I think the topic of the paper is appropriate for OSD. The novelty of the work is apparent, but the my concerns on how this has been done are substantial. Therefore, my overall recommendation is major revision.

Reply: We sincerely thank the Referee for his careful reading of our manuscript and constructive comments and suggestions, which are of great help in improving our study. We have addressed all these comments; our responses are given below.

In this response, the Referee's comments are copied in black, our replies are shown in red, and the following abbreviations are used:

OM - original manuscript,

R1 – Revision #1 - an updated manuscript, which will be submitted as a supplement to this response.

**1) Description of the study site** may be extended by presenting the relative importance of other tidal constituents (S2, K1, O1), e.g. expressed in the value of the form factor F. Why did you consider the M2-tide only? And what is known about the (magnitude of the) tidal currents? This helps interpretation compared to other tidal basins around the world.

Reply: In the "Introduction" of R1, we have added the following sentence to describe the magnitudes of the constituents $S_2$, $K_1$ and $O_1$ relative to $M_2$: "Compared to $M_2$, which has a maximum amplitude of over 2.2 m, the amplitudes of the rest of the constituents are much smaller. The maximum amplitudes of $S_2$, $K_1$ and $O_1$ observed at 11 coastal gauge stations reported by Jan et al. (2004b) are 0.66, 0.39 and 0.27 m,

respectively".

**2) Literature review** should in my opinion be improved in certain respects.

• The large number of references on tides in the Taiwan Strait makes me wonder what has been found in those studies. . .

Reply: Since this study focuses on the tidal dynamics in the strait, we describe mainly the progress in the dynamic aspects without giving a comprehensive review of the progress in the studies of tides in the strait.

• Page 2, Line 12: "was the main component" –> "is the main component".

Reply: Revised as suggested.

• Upon first introduction in Line 18, The extended Taylor method (when using "the", please remove the "'s") requires a reference and an explanation of what 'extended' means here.

Reply: "the extended Taylor's method " in the OM has been replaced with "an extended Taylor method" in R1. (Here, we replace "the" with "an" according to the comment from Referee #2).

• Roos&Velema should in fact be Roos et al (there are more co-authors). Also, unlike suggested by the authors here, the presence of the Dover strait in the south is in fact an open boundary.

Reply: "Roos and Velema" has been changed to "Roos et al.", and the citation in the References is also revised in R1. The statement "all of the studied basins" in the OM is not accurate, and thus "all" is replaced with "most of" in R1.

• I cannot find Table 1 in the .pdf-file that for this review.

Reply: "Roos and Velema, 2011, Table 1" in the OM has been replaced with "Table 1 of Roos et al., 2011" in R1.

• Hendershott & Speranza (Deep Sea Res 1971) is worthwhile mentioning as they followed a similar approach to study the Gulf of California (two Kelvin waves)

Reply: Hendershott & Speranza's paper has been cited in R1.

• Because of the depth-step, one may consider reference to Roos&Schuttelaars(Ocean Dyn 2011)

Reply: Roos & Schuttelaas (2011) has been cited in R1.

• Figure 2: "amphidromic chart" seems better, because it is both co-tidal and corange information that is plotted here. Also: is it Chen and Andersen or Cheng and Andersen?

Reply: We replaced "cotidal" with the more accurate term "tidal system" in R1. "Chen" has been replaced with "Cheng". The reason for not using "amphidromic" is that there is no amphidromic point in the TS, especially in the area shown in Fig. 3.

**3) Model formulation** contains some inaccuracies. First of all, the title of section 2does not really cover the content. I think "Model formulation and solution method" is more appropriate.

Reply: The section title has been revised as suggested.

• Please mention the important simplifications/approximations made here. This is a linear depth-averaged model, the validity of which is relevant. I think this should be discussed at some point.

Reply: According to this comment, we have added the following statements to the

text: "The equations in (1) are two-dimensional linearized shallow water equations on an f-plane with the momentum advection neglected. The equations are the same as those used in the work of Taylor (1922), except that the bottom friction is incorporated, as in Fang and Wang (1966) and Rienecker and Teubner (1980)."
- The pressure gradients in Eq. (1) should have spatial derivatives ($\partial/\partial x$ and $/\partial y$)
Reply: Corrected as suggested.
- Page 3, Line 8: "channel" –> "rectangular channel"
Reply: The word "rectangular" has been added.
- Line 11: "by introducing a collocation method" –> "by applying a collocation method"
Reply: The word "introducing" has been replaced with "applying".
- Page 4, Line 5: "for open rectangular basins" –> "accounting for the finite length of the basin"
Reply: Revised as suggested.
- Line 13: please mention "depth-averaged"
Reply: Added as suggested.
- Line 15: this approximation is known as the f-plane
Reply: The word "f-plane" has been added into the next line.
- Line 16: "cosine wave" is perhaps better rephrased as "monochromatic"
Reply: The word "cosine" has been replaced with "monochromatic".
- Line 17: please put brackets { and } after the real part: Re {(ζ, u, v) exp(iσt)} and please introduce (ζ, u, v) as the complex amplitudes of the quantities introduced previously.
Reply: The brackets have been added.
- Page 5, Line 4: please add "each with a different uniform depth $h_A$ and $h_B$"
Reply: Added as suggested.
- Line 9: would be nice if your radiation condition would include bottom friction. How large us mu typically?
Reply: In the present study we take $\mu = 0.15$ so that $\mu^2$ is of the order of 0.02. If friction is considered, the expression of the radiation condition will become complicated: besides a change in the amplitude ratio, there will be a phase difference between the velocity and elevation. These changes would cause very few differences in the computed results (of the order of $\mu^2$) and are thus ignored in this study.
- Line 23: the formula for wave speed also holds in absence of friction only. . .please reorder
Reply: The sentence has been reordered according to this comment. Here, the formulas hold only under the condition of no friction; if friction is considered, the formulas for wave speed and wave number should be modified {e.g., the wave speed is equal to $\mathrm{Re}\left(\frac{\sigma}{\beta}\right) = \left[\frac{2gh}{\sqrt{1+\mu^2}+1}\right]^{1/2}$, which is equal to $0.9972\sqrt{gh}$ for $\mu = 0.15$}.
I would put the details of Eqs.(9)-(12) and Eqs. (17-24) in an appendix.
Reply: The derivation of these equations has already been made in previous works (e.g., Fang et al., 1991), so it seems unnecessary to give further details in this paper. Furthermore, these equations will be mentioned in the text that follows. Therefore, for

convenience, we wish to retain these equations as they are.
• Line 22: I think it is unnecessary to introduce Q, because you can immediately write
$Q^2 = \alpha^2 - \beta^2$.
Reply: Yes, $Q^2$ is equal to $\alpha^2 - \beta^2$. In R1, $-Q^2$ has been replaced with $-\alpha^2 + \beta^2$ in the expression of $s_n$ (Eq. (16)).

**4) Comparison with observations** is purely visual, which raises some questions. First of all, how did you choose the basin dimensions, orientation? How do you actually project the true geometry, with curved coastlines, onto the rectangular model domain? And, as before: did you consider doing the same for other tidal constituents? Other than that, I find the title of Section 3 confusing, since the model has already been introduced in Section 2. I suggest to change the title into "Application to Taiwan Strait", because that is what is actually done in this section. Also, please avoid if statements when you specify coefficients (Page 7, line 2) an please replace "equal to" with an equality sign =.

Reply: The main purpose of this study is to reveal the dynamics of the $M_2$ wave formation in the strait. We think visual comparison is capable of meeting this goal. No attempt is made to best fit the model results to observations in this study. For the same reason, the basin dimension and orientation, locations of the sidewall and open boundaries have all been chosen through visual inspection. The diurnal constituents in the strait are small and have a simple structure (please see the following tidal system chart of the largest diurnal constituent $K_1$). Thus, we are not interested in the study of their dynamics. The dynamics of $S_2$ is the same as that of $M_2$, so we just pay attention to $M_2$. The title of Section 2 has been changed according to this comment in R1. "If" has been replaced with "In this study"; "equal to" has been replaced with the sign "=".

[Figure]

Fig. 1.1. K1 tidal system in the Taiwan Strait and its neighbouring area, (a) amplitudes in cm (b) phase-lags in degrees (from Zhu et al., 2009).
(In the figure number, the first "1" represents "Author's Response to Referee #1")

**5) Interpretation of role of Kelvin and Poincare modes** can readily be deepened by further analysis. First of all, what is the wavelength of the Kelvin waves? (I see it is mentioned later on but already here it is relevant). And are the Poincare modes free or bound (from the depth and width values I guess they are all bound), and what is the typical length scale of decay of the lowest Poincare mode? This gives insight in the extent to which these modes affect the amphidromic pattern in the (interior of the)

Taiwan Strait.

Reply: In the first paragraph of Section 3.1, we have added the following statement: "From these parameter values, we can obtain the wavelength of the M2 Kelvin wave as 1009 km. Since the basin width is smaller than half of the Kelvin wavelength, the Poincaré modes can only exist in a bound form (Godin, 1965; Fang and Wang, 1966). The e-folding length of decay of the lowest Poincaré mode is approximately 63 km, that is, the amplitude of this mode reduces to approximately 37% relative to its maximum value at a distance of 63 km away from the boundary. Equivalently, it may also reduce to approximately 20% relative to its maximum value at a distance of 100 km. The length scales of decay for higher order Poincaré modes are even shorter."

Also, I do not understand the statement that frictional force would be a major factor (as mentioned here and repeated in Section 5). I think this is not the case, in view of the mild amplitudes and large depths. Can you support this statement? I suspect you would still get a good fit if bottom friction were switched off.

Reply: Here, the words "the Coriolis and frictional forces" have been replaced with "the Coriolis force and the weaker northward wave" according to this comment.

• Page 7, Line 23: "inclusion of the Poincaré modes improves"

Reply: Revised as suggested.

• Page 8, Line 10: Also possible is that the assumption of uniform depth is too restrictive in this Taylor approach. . .

Reply: We agree with this point of view and have therefore added "at a uniform depth" following "Kelvin wave", such that the statement is now "This amplitude variation cannot be completely represented by the superposed Kelvin wave at a uniform depth".

• Page 8, Line 21: this is a basic statement about progressive waves and therefore not really insightful in my opinion.

Reply: The words "due to propagation direction" have been deleted in R1.

• Page 10, I do not understand the statement on resonance. This may hold for closed basins, but here we have a topographic step. . .

Reply: Resonance is also possible in a basin with a topographic step. This can be illustrated with one-dimensional problems as follows:

For a basin of uniform depth $h$ and length $L$, if it has a closed end at x=0 and an

opening at x=L, where tidal elevation is given as $\zeta(L,t) = a\cos \sigma t$, then the

elevation in the basin is $\zeta(x,t) = a\frac{\cos kx}{\cos kL}\cos \sigma t$, where $k = \sigma/\sqrt{gh}$. Resonance

will occur if $kL = \frac{\pi}{2}, \frac{3\pi}{2}, ....$ Please see Godin, 1993 (continental Shelf Res., 13(1), p. 103).

For a shallow basin of uniform depth $h$ and length $L$, if its mouth is at x=L, where tidal elevation is given as $\zeta(L,t) = a\cos\sigma t$, and it has a topography step at x=0, which connects with a deep basin of infinite-depth, then the elevation in the shallow

basin is $\zeta(x,t) = a\frac{\sin kx}{\sin kL}\cos\sigma t$, where $k = \sigma/\sqrt{gh}$. Resonance will occur if

$kL = \pi, 2\pi, ....$ Please see Jan et al, 2002 (Journal of Oceanography, 58, p. 849).

**6) Phrasing** in general should be more precise in my opinion. For example, avoid the unnecessary and confusing use of the verb "can". My suggestion is to consult a native speaker of the English language with knowledge of the topic to revise the text. Here I explain what I mean by giving some suggestions to improve the abstract (line 8-22).

Reply: R1 has been edited for English by a native English speaker from a language service company. Please see the following certificate issued by them

[Figure]

**EDITORIAL CERTIFICATE**

This document certifies that the manuscript listed below was edited for proper English language, grammar, punctuation, spelling, and overall style by one or more of the highly qualified native English speaking editors at American Journal Experts.

**Manuscript title:**
An analytical study on M2 tidal wave in the Taiwan Strait with an extended Taylor method

**Authors:**
Di Wu, Guohong Fang, Xinmei Cui, Fei Teng

**Date Issued:**
October 23, 2017

**Certificate Verification Key:**
A75B-6EB6-9F56-B486-FF3A

[Figure]

This certificate may be verified at www.aje.com/certificate. This document certifies that the manuscript listed above was edited for proper English language, grammar, punctuation, spelling, and overall style by one or more of the highly qualified native English speaking editors at American Journal Experts. Neither the research content nor the authors' intentions were altered in any way during the editing process. Documents receiving this certification should be English-ready for publication; however, the author has the ability to accept or reject our suggestions and changes. To verify the final AJE edited version, please visit our verification page. If you have any questions or concerns about this edited document, please contact American Journal Experts at support@aje.com.

American Journal Experts provides a range of editing, translation and manuscript services for researchers and publishers around the world. Our top-quality PhD editors are all native English speakers from America's top universities. Our editors come from nearly every research field and possess the highest qualifications to edit research manuscripts written by non-native English speakers. For more information about our company, services and partner discounts, please visit www.aje.com.

• Page 1, Line 8: "M2" –> "semidiurnal lunar (M2)"
Reply: Revised as suggested.
• Line 8, "The extended Taylor's method", remove "'s": and is it sufficiently clear what this means?
Reply: Revised as suggested.
• Line 10, "but" –> "and" (because this does not really signify a contradiction!)
Reply: Revised as suggested.
• Line 10: "friction forces" –> "bottom friction"
Reply: Revised as suggested.
• Line 16: "can further improve" is unclear. Better: "Inclusion of Poincaré modes further improves"
Reply: Revised as suggested.
• Line 18: "can be reflected" –> "is reflected" (I guess this is what you mean)
Reply: Revised as suggested.
• Line 21: same with "can" as in Line 18.

Reply: Revised as suggested.

**Anonymous Referee #2**

General comments

This paper contains original contribution to analytical tide modeling using an Taylor' method. Although there are quite many thing to be clarified and improved, it is believed that authors can revise the manuscript without much difficulties. This paper is therefore recommended for the publication in OS with minor corrections.

Reply: We sincerely thank the Referee for his careful reading of our manuscript, as well as the constructive comments and suggestions which are of great help for improving our study. We have addressed all these comments; our responses are given below.

   In this response, the Referee's comments are copied in black, our replies are shown in red, and the following abbreviations are used:

OM - original manuscript,

R1 – Revision #1 - an updated manuscript, which will be submitted as a supplement to this response.

Specific comments

Title

Pg.1, lines 1-2 Better to replace "the extended Taylor's method" to "an extended Taylor's method".

Reply: The expression "the extended Taylor's method" has been replaced with "an extended Taylor method" in R1. Here, we have also removed "s" according to the suggestion of Referee #1.

Abstract

Pg. 1, line 8: Again use "an extended Taylor's method Pg. 1, lines 21-22: The sentences are a little bit unnatural. Include how much the northward KW is strengthened, that is, quantitatively, saying it is of secondary importance.

Reply: Here, the expression "the extended Taylor's method" has also been replaced with "an extended Taylor method". Since the results of Ex. 3 (with the Luzon Strait input being considered) contain some uncertainties (please see discussion in Section 5) we would not give a quantitative estimate in Abstract, but just state that "the forcing is thus of some (but lesser) importance to the $M_2$ tide in the TS."

1 Introduction

Pg. 2, line 4: The expression "anti-nodal band" is not familiar. Is there anyone to use the expression?

Reply: We call the area where the vertical movement of an oscillating wave is greatest the "anti-nodal band". The word "antinode" can be found, for example, in Figure 5:3 of the monograph *Tides, Surges and Mean Sea-Level*, by David T. Pugh, 1987. (In the

USA it is called a "loop"; please see page 14 of *Tide and Current Glossary* by the Center for Operational Oceanographic Products and Services, NOAA. Since OS is a European journal, we have followed Pugh's usage).

Pg. 2, line 18: Again use "an extended Taylor's method

Reply: The phrase "the extended Taylor's method" here has also been replaced with "an extended Taylor method".

Pg. 2, lines 27-28: What is the basis of "The statement "the topographic step south of the TS acts as a permeable interface which can only partially reflect the incident wave and ....by nearly 180 degree at the step". Previous studies? If not, authors already assume partial reflection of northward and southward waves at the step. More careful writing is needed.

Reply: Yes, this is a result of previous studies. In R1, we have added a citation for Dean and Dalrymple (1984, Section 5.5) as a reference to the reader.

Pg. 4, line 9: Again use "an extended Taylor's method.

Reply: The phrase "the extended Taylor's method" here has also been replaced with "an extended Taylor method".

3. An analytical model for the Taiwan Strait

3.1 Model configuration and solution

Pg. 7, lines 1-2: Reference for the friction coefficient formula is required.

Reply: We have cited "e.g., Chapter 8 of Dronkers, 1964" here and added "Dronkers, J. J.: *Tidal Computations in Rivers and Coastal Waters*, North-Holland Publishing Company, Amsterdam, 518 pp, 1964" to the reference section of R1.

Pg. 7, line 5: The expression "observed harmonic constants from the global tide model" is a little bit strange. It is computed results not observed values. Recommend to change the expression.

Reply: Since this global model is not numerically simulated but is based on satellite observations, we regard the model values as observations. For clarity, we have added the following sentence to the Introduction of R1: "Figure 2 displays the distribution of the M2 tidal constituent based on the global tidal model DTU10, which is constructed on the basis of multi-mission altimeter observations. Hereafter, we shall regard the DTU10 model results as observations." It seems that the words "observation-based" are more accurate than "observed", but we feel the former is too lengthy.

Pg. 7, line 6: In Figure 3(a) open boundary values appear to be somewhat different with Figure 2. Describe how the global model results were adopted to the analytical model.

Reply: The open boundary values are derived from the DTU10 model values through linear interpolation. The impression of the difference mentioned in this comment may be caused by the following two issues. (1) The rectangle shown in Fig. 1 in the OM is a sketch diagram and is not accurate. Now Fig. 1 has been replaced in R1 with a new figure, in which the rectangle is plotted at its exact location. (2) The other cause might be that we use different contour intervals for Fig. 2 and Fig. 3. To eliminate this impression, we redrew Fig. 2 with a finer interval (10 degrees) for phase-lags, as shown below.

[Figure]

Fig. 2.1 The M$_2$ tidal system in the Taiwan Strait and its neighbouring area. This figure is the same as Fig. 2 of our manuscript but uses a finer contour interval (10 degrees) for phase-lags.
(In the figure number, the first "2" represents the "Author's Response to Referee #2".)

Pg. 7, lines 15-16: The statement "that is, the wave propagates southward in the northeast area and propagate northward in the southeast area" may be incorrect in strict sense. Figure 4 shows in whole area there are southward and northward Kelvin waves. Rewriting is needed.
Reply: This statement refers to the tidal patterns given in Fig. 3. The propagation directions are shown with arrows in the following figure.

[Figure]

Fig. 2.2 Propagation directions of M2 tide in the north-eastern and south-eastern areas of the Taiwan Strait.
(In the figure number, the first "2" represents the "Author's Response to Referee #2".)

3.2 Kelvin waves and Poincare modes

Pg. 8, line 6: In Figure 3(a) open boundary values appear to be somewhat different with Figure 2. Describe how the global model results were adopted to the analytical model.

Reply: Please see the reply to the comment on Pg. 7, line 6 above.

Pg. 8, lines 9-10: The statement "the amplitude variation along the northern boundary ......owing to the fact that the M2 tide is from the Pacific Ocean" is obscure. The northern boundary values of Fig. 3c and Fig.3a can be different from each other even though the M2 tide did not come from Pacific Ocean. More careful discussions are required.

Reply: To make the description clearer, the statement "This is owing to the fact that the $M_2$ tide is from the Pacific Ocean, its amplitude increases from the deeper outer shelf toward the shallower inner shelf" in the OM has been replaced with the statement "This shows that near the boundary, the Poincaré modes are of a certain importance. The existence of the Poincaré modes is related to the fact that the M2 tide is from the Pacific Ocean; its amplitude increases from the deeper outer shelf toward the shallower inner shelf." in R1.

Pg.8, lines 17-18: The statement "it is not a controlling factor" is too much definite. Obliqueness might be partly effective. Improve the statement.

Reply: We have changed "it is" to "it seems".

4 Formation mechanism of the northward Kelvin wave in the Taiwan Strait

4.1 Reflection of the incident wave from the East China Sea at the topographic step

Pg. 9, line 17: "observation taken from the global tide model" needs to be changed as mentioned earlier.

Reply: The design of experiments 1-3 has been changed according to your comment on Pg. 14, line 1-2; these words have been deleted in R1. Please see the response to your comment on Pg. 14, line 1-2 below.

Pg. 10. 4: Include reference for "the resonant period of the ECS(13.7h)".

Reply: We have added ", obtained by Cui et al., 2015" after "13.7 h".

5. Summary and discussion

Pg. 14, lines 1-2: Regarding the statement "the reflected wave is slightly weaker", authors implied that there is a partial reflection of southward KW at the step. It is noted that at the northern open boundary southward KW, northward KW and Poincare waves are all specified in Ex.1. If there is northward KW at the northern open boundary, there should be northward KW at the southern open boundary, regardless of topography. It is curious to know what will happen if only southward KW is imposed at the northern boundary in Ex.1. Additional experiment is recommended to clarify the partial reflection at the step. You may include results in appropriate place.

Reply: This is an important suggestion. According this comment, we have redesigned our experiments and replaced Figs. 5, 6, 7 and the related discussion with these new results. In particular, the statement describing the original experiments "As in the single basin solution, the open boundary condition (7) is used at the northern opening with values of $\hat{\zeta}$ equal to those taken from the global tidal model DTU10" in the OM has been replaced with the statement describing the redesigned experiments "The experimental design for area A is similar to that of Roos and Schuttelaars (2011): a southward Kelvin wave is specified to be identical to the single basin solution, as shown in Fig. 4a in the preceding section. The Poincaré modes trapped at the cross section x = 0 are neglected, while those trapped at the cross section x = 400 km are retained." in R1.

Technical corrections

Pg. 1, line 29: Better to replace "M2 amplitudes" to "M2 tide".

Reply: We already use "tide" for the subject of this sentence, so it does not seem appropriate to use "tide" again.

Pg. 3, lines 5-6: Definition of phase lag needs to be added.

Reply: In R1, we have added "Solid lines represent Greenwich phase-lag (in degrees)" into this figure caption.

Pg. 11, line 2: In Figure caption, there is no (d).

Reply: Amended: "(d)" has been added.

**Modification list**

(According to Revision#1-Updated manuscript)

**Page 1.**

Line 3. The title has been replaced with "An analytical study of $M_2$ tidal waves in the Taiwan Strait using an extended Taylor method".

Line 10. "are featured by" has been changed to "feature", and "$M_2$" has been changed to "semidiurnal lunar ($M_2$)".

Line 10. (and other place: Page2. Line30) "The extended Taylor's method" has been changed to "An extended Taylor method".

Line 12. "but the Coriolis and bottom friction force" has been changed to "and the Coriolis force and bottom friction forces".

Line 16. (and several other places: Lines 18, 21, 24; Page10. Lines 6, 9, 17; Page17. Lines 17, 18; Page18 Line 6) The word "can" has been deleted.

Line 18. "The superposition of Poincaré modes can further improve" has been replaced with "Inclusion of Poincaré modes further improves".

Line 19. (and other place: Page7. Lines 13) "In order to" has been replaced with "To".

Line 23. "The forcing at the Luzon Strait" has been changed to "Inclusion of the forcing"

Line 24. "and thus is of secondary importance to the $M_2$ tide in the TS" has been changed to "and the forcing is thus of some (but lesser) importance to the $M_2$ tide in the TS"

Line 28. (and several other places: Page2. Lines 1, 6; Page 10. Lines 8, 9, 12; Page 11.  Lines 3, 4; Page 12. Lines 24, 27; Page 13. Line 4; Page 16. Line 5) The word "about" has been replaced with "approximately".

Line 28. "mostly located" has been changed by "located mostly".

**Page 2.**

Line 4. "The greatest amplitude by tidal gauge observation…" has been changed to "The greatest amplitude based on tidal gauge observations…".

Line 5. "… is 1.73 m at Taichung, while that along the mainland coast is 2.10 m at Matsu" has been changed to "… is 1.73 m at Taichungt and is 2.10 m at Matsu near the mainland coast".

Lines 6-8. "… 20 km away from the coast,the satellite observation indicate that the greatest amplitude, exceeding 2.2 m, appears near Haitan Island, located south of Matsu (Fig. 2)" has been changed to "… 20 km away from the coast. Satellite observations indicate that the greatest amplitude appears near Haitan Island, located south of Matsu (Fig. 2), and exceeds 2.2 m".

Lines 10-17. "which is called the asymmetry by Yu et al. (2015). The tides in the TS have attracted a great number of studies since 1980s" has been deleted, and "and this feature is called asymmetry by Yu et al. (2015). Compared to $M_2$, which has maximum amplitude over 2.2 m, the amplitudes of the rest of the constituents are much smaller: the maximum amplitudes of $S_2$, $K_1$ and $O_1$ observed at 11 coastal gauge stations reported by Jan et al. (2004b) are 0.66, 0.39 and 0.27 m, respectively. Figure 2 displays the distribution of the M2 tidal constituent based on the global tidal model DTU10, which is constructed on the basis of multi-mission altimeter observations. Hereafter, we shall regard the DTU10 model results as observations. The tides in the TS have attracted a great number of studies since the 1980s. Most studies have attempted to establish accurate numerical models and thus to give accurate spatial structures of the tides and tidal currents in the strait." has been added.

Line 19. "Most investigators have" has been changed to "It has been well".

Lines 24, 25. The word "was" has been changed to "is".

Line 24. The word "that" has been added.

Line 25. The word "made" has been replaced with "completed", and the word "on" has been replaced with "of".

Line 26. The word "a" has been added before "special focus on …".

Line 27. The word "modeling" has been replaced with "modelling".

Line 28. The word "in" before "Jan et al. (2002)" has been replaced with "in".

Line 30. "(see Section 2 for details)" has been added after "an extended Taylor method".

Line 31. "… in the natural basin, and thus enable." has been changed to "… in the natural basin. This enables".

**Page 3.**

Line 4. (and several other places: Page5. Lines 4, 8) "Taylor's problem" has been replaced by "The Taylor problem"

Line 4. "(Hendershott and Speranza, 1971)" has been added after "…tidal dynamic problem".

Lines 5-6. "(e. g., Roos and Velema , 2011, Table 1)" has been changed to "(e. g., Table 1 of Roos and Velema,et al., 2011)".

Line 6. "all the studied" has been changed to ", most of the studied".

Line 7. ", thus" has been added after "…the incident tidal wave".

Line 8. The word "which" has been changed to "that".

Line 9. "(see Section 5.5 of Dean and Dalrymple, 1984)" has been added.

Line 10. "the Taylor's method" has been replaced by "Taylor's method".

Line 11. Figure 1. has been replaced with a new figure.

**Page 4.**

Line 5. (and other places: Page 9. Line5) "Cotidal chart of M2 constituent" has been changed to "The M2 tidal system", and "and its neighbouring area" has been added after "Taiwan Strait".

Lines 5-6. "Chen and Andersen, 2011" has been corrected by "Cheng and Andersen, 2011", and the sentence "Solid lines represent the Greenwich phase-lag (in degrees), and dashed lines represent amplitude (in metres)." has been added.

Line 8. The title "Solution Method" has been changed to "Model formulation and solution method".

Line 9. The word "rectangular" has been add before "channel of uniform depth".

**Page 5.**

Line 1. "Defant in 1925" has been changed to "In 1925, Defant", and the word "introducing" has been replaced with "applying".

Line 2. "version of" has been added before "Taylor's problem", and "Defant' approach" has been corrected to "Defant's approach".

Line 4. "in the governing equations" has been added.

Line 6. "towards" has been changed to "towards".

Lines 7. The sentence "The mechanism of the shift of the amphidromic point was also explained by Hendershott and Speranza (1971), in which the dissipation was assumed to occur at the closed end of the basin rather than during the wave propagation." has been added.

Line 9. "…enabling solutions for open rectangular basins" has been changed to "enabling solutions accounting for the finite length of the basin".

Line 10. (and other place: Line 14) "Roos and Velema (2011)" has been corrected to "Roos et al. (2011)", and ", Roos and Schuttelaars (2011)" has been added.

Line 17. The equations have been corrected to:

$$\begin{cases} (\mu + \mathrm{i})u - \nu v = -\dfrac{g}{\sigma}\dfrac{\partial \zeta}{\partial x} \\ (\mu + \mathrm{i})v + \nu u = -\dfrac{g}{\sigma}\dfrac{\partial \zeta}{\partial y} \\ \zeta = \dfrac{\mathrm{i}h}{\sigma}\left[\dfrac{\partial u}{\partial x} + \dfrac{\partial v}{\partial y}\right] \end{cases}$$

Lines 18-21. The sentence has been changed to "where $t$ represents time; $(x, y)$ are the Cartesian coordinates; $(\tilde{u}, \tilde{v})$ are the depth-averaged velocity components in the $(x, y)$ directions; $\tilde{\zeta}$ is the tidal elevation; $h$ is the water depth, assumed uniform; $\gamma$ is the frictional coefficient, taken as a constant; $g = 9.8\,\mathrm{ms^{-2}}$ is the acceleration due to gravity; and $f$ is the Coriolis parameter, also taken as a constant due to the smallness of the study area."

Lines 21-24. The sentence "The equations in (1) are two-dimensional linearized shallow water equations on an $f$-plane with the momentum advection neglected. The equations are the same as those used in the work of Taylor (1922), except that the bottom friction is incorporated, as in Fang and Wang (1966) and Rienecker and Teubner (1980). When a monochromatic …" has been added.

Line 24. (and several other places: Line 27; Page 6 Line5; Page 7 Line 14) "as follows:" has been added.

Line 25. The equation $(\tilde{\zeta}, \tilde{u}, \tilde{v}) = \mathrm{Re}(\zeta, u, v)e^{\mathrm{i}\sigma t}$ has been corrected to $(\tilde{\zeta}, \tilde{u}, \tilde{v}) = \mathrm{Re}\{(\zeta, u, v)e^{\mathrm{i}\sigma t}\}$.

Lines 26-27. The sentence has been changed to "where $(\zeta, u, v)$ are complex amplitudes of $(\tilde{\zeta}, \tilde{u}, \tilde{v})$, respectively, and $\sigma$ is the angular frequency of the wave, $\mathrm{i} \equiv \sqrt{-1}$. For this wave, the equations in (1) reduce as follows:" has been added.

**Page 6.**

Line 1. The equations (3) have been corrected to :

$$\begin{cases} (\mu + \mathrm{i})u - \nu v = -\dfrac{g}{\sigma}\dfrac{\partial \zeta}{\partial x} \\ (\mu + \mathrm{i})v + \nu u = -\dfrac{g}{\sigma}\dfrac{\partial \zeta}{\partial y} \\ \zeta = \dfrac{\mathrm{i}h}{\sigma}\left[\dfrac{\partial u}{\partial x} + \dfrac{\partial v}{\partial y}\right] \end{cases}$$

Line 4. "Consider" has been changed to "Considering", the word "of" before $B$ has been deleted, and "put" has been changed to "placed".

Line 5. The word "another" has been changed to "the other".

Line 14. ", each with a different uniform depth of $h_A$ and $h_B$" has been added.

Line 16. "Eqs. (8) are …" has been replaced with "The equations in (8) show the …".

Line 17. The word "on" has been changed to "for".

Lines 16-19. Some ", " has been added.

Line 20. The word "of" has been deleted.

Line 22. The word "in" has been added before "(3)", and the word "of" has been added before "(4)".

Line 23. The typeface of "β" has been change to Italic.

Lines 23-24. The sentence "Note: the error occurred during their preparation of the manuscript and the correct expression was used in their computations." has been added.

**Page 7.**

Line 10. The equation (16) has been changed to :

$$s_n = (r_n^2 - \alpha^2 + \beta^2)^{\frac{1}{2}}$$

Line 11. "$c = \sqrt{gh}$" has been changed to "with $c = \sqrt{gh}$ being the wave speed", and "the" has been added before "absence".

Line 12. The equation about $Q^2$ has been deleted, and "for each n " has been added.

Line 17. The equation (18) has been changed to:

$$B_n = \frac{\nu(\mu+i)(\alpha^2-\beta^2)}{(\mu+i)^2 r_n^2 + \nu^2 s_n^2}$$

Line 24 "$(a,b,\kappa_n,\lambda_n)$ are Coefficients related" has been changed to "Coefficients $(a,b,\kappa_n,\lambda_n)$ are related".

**Page 8.**

Line 2. The word "in" has been changed to "when".

Line 4. ", and" has been added before "the total number of …".

Line 9. The word "fast" has been changed to "quickly".

Line 11. The title "An analytical model for the Taiwan Strait" Application to the Taiwan Strait".

Line 16. The word "put" has been changed to "place", and the word "the" has been added before "x axies …" and "y axies …".

Line 17. "an" has been added `before "offshore direction", and "by" has been changed to "to".

Line 18. "But for short" has been changed to "However, to keep it short,", and "by" has been changed to "to".

Line 19. (and other place: Page 12. Line 13) "is taken" has been changed to "is taken as".

Lines 20, 2. The typeface of "γ" has been change to Italic.

Lines 24-29. The sentence "In this study, we take $C_D$ =0.0026 and $U$ =0.5 m/s based on the numerical results of Fang et al. (1984), and then, $\mu = \gamma/\sigma$ is approximately equal to 0.15. From these parameter values, we can obtain the wavelength of the $M_2$ Kelvin wave as 1009 km. Since the basin width is smaller than half of the Kelvin wavelength, the Poincaré modes can only exist in a bound form (Godin, 1965; Fang and Wang, 1966). The e-folding length of decay of the lowest Poincaré mode is approximately 63 km, that is, the amplitude of this mode reduces to approximately 37% relative to its maximum value at a distance of 63 km away from the boundary. Equivalently, it may also reduce to approximately 20% relative to its maximum value at a distance of 100 km. The length scales of decay for higher order Poincaré modes are even shorter." has been added.

Line 30 "In this study, the" has been added before "families of Poincaré modes…", and "are" has been deleted.

**Page 9.**

Line 3. Figure 3. has been replaced with a new figure.

Line 5. (and other places: Line 8; Page 10. Line 4) The word "cotidal" has been changed to "tidal system".

Line 5. "(in degrees)" has been added.

Line 8. (and other places: Page 12. Lines 24, 27; Page 17. Line 4) "the" has been added before "$M_2$".

Line 9. "It can be seen that although" has been changed to "Although".

Line 11. "…has the features that the amplitudes are …" has been changed to "…features …", and "amplitudes" has been added after "greater".

Line 14. "…, wave nature, that …" has been changed to "… wave nature,. That …".

Line 15. "A highest amplitude band is present …" has been changed to "The highest amplitude band present …", and "…, appearing    …" has been changed to "…, and appears    …".

**Page 10.**

Line 3. "the" has been added before "Poincaré modes", and "It can be observed that the contribution of the Poincaré modes is much smaller than that of Kelvin waves" has been changed to "The contribution of the Poincaré modes is observed to be much smaller than that of the Kelvin waves".

Line 6. "inclusion of the" has been added before "Poincaré modes".

Line 10. "… with southward one stronger then the northward one" has been changed to "…, with the southward one being stronger than the one moving northward".

Lines 10-11. "Here both the Coriolis and frictional forces …" has been changed to "Here, both the Coriolis force and the weaker northward wave …".

Line 14. "to enhance" has been changed to "enhances".

Line 18. "This is owing …" has been replaced with "This shows that near the boundary, the Poincaré modes are of a certain importance. The existence of the Poincaré modes is related …".

Line 21. "at a uniform depth" has been added, and "for" has been added before "their difference".

Line 27. "but it is not a controlling factor" has been changed to "but it seems not to be a controlling factor".

Line 28. "the" has been added before "southward and northward Kelvin waves".

Line 31. "due to propagation direction" has been deleted, and the word "the " has been added before.

**Page 11.**

Line 1. The word "the" has been added before "opposite".

Line 4. (and other place: Line 9) . The word "them" has been changed to "the waves".

Lines 17-19. The sentence "In the preceding section, we have shown that the northward Kelvin wave is weaker than the northward wave on average, but they have a similar magnitude along the Taiwan coast. In this section, we will examine the formation mechanism of the northward Kelvin wave." has been added.

Line 19. "the" has been added before "reflection".

Line 20. "and " has been added before "another is an incident wave …".

**Page 12.**

Lines 6-10. "As in the single basin solution, the open boundary condition (7) is used at the northern opening with values of equal to the observations taken from the global tidal model DTU10." has been changed to "The experimental design for area A is similar to that of Roos and Schuttelaars (2011): a southward Kelvin wave is specified to be identical to the single basin solution, as shown in Fig. 4a in the preceding section. The Poincaré modes trapped at the cross section $x =0$ are neglected, while those trapped at the cross section $x =400$ km are retained.".

Lines 13, 14. "Fig. 3a" has been changed to "Fig. 3c".

Line 20. The word "made" has been changed to "performed".

Line 21. "…, meaning …" has been changed to "…, obtained by Cui et al., 2015). This means …".

Line 29. "south" has been changed to "southern", and "the" has been added before "single area case".

Line 32. "and the" has been added before "reflected and transmitted Kelvin waves".

**Page 13.**

Line 2. The number "0.63" has been changed to "0.64"

Line 3. The numbers "0.60" and "0.38" have been changed to "0.61" and "0.37", respectively.

Lines 4-6. The typeface of variables have been corrected to Italic.

Line 7. "being" has been added before "taken into account…" and "ignored".

Line 8. Figure 5. has been replaced with a new figure.

**Page 14.**

Line 2. "(d)" and "(in degrees)" has been add.

Line 6. "carry out" has been changed to "performed".

Line 6. "the deep basin is" has been changed to ", the deep basin has".

Line 7. "… radiate freely southward …" has been changed to "… to freely radiate southward …".

Line 9. "… the deep basin is …" has been changed to "… the deep basin has …".

**Page 15.**

Line 1. Figure 6. has been replaced with a new figure.

Line 5. The word "making" has been changed to "making".

Lines 6, 7. "the Taylor's model" has been changed to "Taylor's mode".

Line 7. "… not allow to open any part of sidewalls" has been changed to "… not allow any part of the sidewalls to open".

Line 8. The word "simulate" has been changed to "solve", and "still" has been deleted.

**Page 16.**

Lines 3-5. Some "," have been added.

Line 4. "about" after "roughly" has been deleted.

Line 5. "Since a significant part of incident wave …" has been changed to "Since a significant portion of the incident wave …".

Line 9. The word "southeastern" has been changed to "south-eastern".

Line 10. Figure 7. has been replaced with a new figure.

**Page 17.**

Line 4. "In the present study we first establish an analytical model …" has been changed to "In the present study, we first stablished an analytical model …".

Line 8. "carry" has been changed to "carried".

Line 10. "From this study we obtain the following results" has been changed to "From this study, we have obtained the following results".

Line 11. "by" has been added after "basically represented".

Line 12. "being" has been added before "stronger than the latter".

Line 15. The word "composition" has been changed to "superposition".

Line 16. "the Coriolis force and friction …" has been changed to "the Coriolis force and the weaker northward wave …".

Line 16. "Inclusion of the Poincaré modes into the analytical model can improve the model result: …" has been changed to "Inclusion of the Poincaré modes in the analytical model improves the model results: …".

**Page 18.**
Line 1. The word "have" has been changed to "make".
Line 4. "…in the strait, though …." has been changed to "… in the strait. However, …".
Line 5. "the" has been added before "observations".
Line 7. "TS tides" has been changed to "M2 tide in the TS".
Line 11. "the water depths change from …" has been changed to "the water depth changes from …".
Line 12. "… of continental slope there" has been changed to "… of the continental slope at that location".
Lines 13-14. "…in the result for the magnitude of reflected wave" has been changed to "…in the results for the magnitude of the reflected wave".
Line 17. "…,the National Natural Science Foundation of China (Grant No. 41706031)" has been added.
Lines 19-20. "The authors sincerely thank Dr. Huthnance and two 20 anonymous Referees for their constructive comments and suggestions, which are of great help in improving our study" has been added.
Lines 29-30. "Dronkers, J. J.: Tidal Computations in Rivers and Coastal Waters, North-Holland Publishing Company, Amsterdam, 518 pp, 1964." has been added.

**Page 19.**
Line 1. "… , 8, 60-77" has been added.
Line 8. "Bruce, P" has been changed to "Paker, B", and "ins." has been corrected to "inc.".
Lines 12-13. "Hendershott, M. C., and Speranza, A.: Co-oscillating tides in long, narrow bays; the Taylor problem revisited. Deep-Sea Research, 18, 959–980, 1971." has been added.
Line 15. "212502" has been changed to "21".
Line 31. "Roos P. C., and Schuttelaars H. M.: Influence of topography on tide propagation and amplification in semi-enclosed basins, Ocean Dynamics, 61, 21-38, 2011." has been added.

**Page 20.**
Line 8. "Roos, P. C., Velema, J. J." has been changed to "Roos, P. C., Velema, J. J., Hulscher, S. J. M. H., and Stolk, A.".
Line 8. "… 39-450, 1985 (in Chinese with English abstract)." has been changed to "… 39-450 (in Chinese with English abstract), 1985.".

Revision #1 – Updated manuscript

[revised manuscript text omitted]

---

## Author Response (AR2)

**Response to comments of Topic Editor**

**Topic Editor Decision: Publish subject to minor revisions (review by editor)** (27 Nov 2017) by John M. Huthnance

Comments to the Author:

Dear Authors

Thank-you again for your revised manuscript. The referee has now seen it and asks for one correction as well as making two remarks which you should perhaps also take into account.

Reply:

Dear Dr. Huthnance:

Thank you very much for your decision on our manuscript and making invaluable comments and suggestions! Our response to your and Referee's comments is given in red below. In this response we use the following abbreviations:

R1 - The previous revised manuscript,

R2 - The present revised manuscript,

TPR - The present response.

Correction: "I think the statement on basin width being smaller than half the Kelvin wavelength leading to bound Poincaré waves is not correct. The expression for critical basin width is in fact more complicated and depending on Coriolis parameter as well: bcrit=pi*sqrt(gh/[omega^2-f^2]) as one may derive e.g. from Eq.(8) of Hendershott & Speranza [1971] or other sources. Please revisit this statement and correct."

Remarks:

Reply: We thank Referee #1 for bringing our attention to Eq. (8) of Hendershott and Speranza (1971, hereafter HS71). The formula for critical basin width given in this comment is surely correct; but our statement "Since the basin width is smaller than half of the Kelvin wavelength, the Poincaré modes can only exist in a bound form" on page 7 is also correct. The reason is as follows. The formula given in this comment is

$$B < \pi \sqrt{\frac{gh}{\sigma^2 - f^2}} \tag{R1}$$

in which we use the same notation as used in our manuscript for consistence; the

equality sign in the comment is replaced with inequality sign in (R1) following previous convention (including HS71 and Fang and Wang (1966) which we shall mention again later). The above inequality can be rewritten in the form

$$B < \frac{\lambda}{2} \sqrt{\frac{1}{1-(\frac{f}{\sigma})^2}} \tag{R2}$$

where $\lambda = 2\pi \frac{\sqrt{gh}}{\sigma}$ is the Kelvin wavelength. Our statement "the basin width is smaller than half of the Kelvin wavelength" can be mathematically expressed as

$$B < \frac{\lambda}{2} \tag{R3}$$

It can be readily seen that as long as (R3) is satisfied, the inequality (R2) is also satisfied, because $\sqrt{\frac{1}{1-(\frac{f}{\sigma})^2}}$ is always equal to or greater than 1. Therefore, our statement "the Poincaré modes can only exist in a bound form" has no problem.

**Additional remarks:**

(a) In his previous comments Referee #1 proposed an equation $Q^2 = \alpha^2 - \beta^2$. We are sorry for having not found the error in his formula, and adopting this formula in Eq. (16) of R1. The correct equation should be $Q^2 = \beta^2 - \alpha^2$. Therefore, In R2 we have changed Eq. (16) to

$$s_n = (r_n^2 + \alpha^2 - \beta^2)^{\frac{1}{2}} \tag{R4}$$

(b) Five years before the work of HS71, Fang and Wang (1966, hereafter FW66) had already given the inequality (1) in a slightly different form. The inequality (47) of their paper is as follows:

$$\frac{1-s^2}{l^2} < 1 \tag{R5}$$

in which s and $l$ are dimensionless quantities, defined as (their Eq. (6))

$$s = f/\sigma \ ; \ l = \frac{\pi}{B} \frac{\sqrt{gh}}{\sigma} \tag{R6}$$

Inserting (R6) into (R5), taking square root of the result, and multiplying the root by $\frac{\pi\sqrt{gh}}{\sigma} \sqrt{\frac{1}{1-(\frac{f}{\sigma})^2}}$ , these immediately yield (R1). In (R6) $l$ is in fact equal to $\frac{\lambda/2}{B}$, so that from (R5) and (R6) we can readily get the inequality (R2). The paper FW66 was published more than 50 years ago in a Chinese journal, and now is not internationally available, we shall attach its e-version (PDF) as a supplement to this response or to our

manuscript.

(c) The restriction of the inequality (R3) is stronger than (R2), but (R3) is simpler and easier to be remembered. Furthermore, it is of some historical significance: this condition was regarded as the condition for a narrow sea by Grace in 1931 (please see the inequality (6) of Godin, 1965). This condition was also presented in FW66 (the "narrow sea theory" was first developed by Sterneck in 1922).

(d) If $f > \sigma$, the inequality (R1) as well as (R2) becomes meaningless. We can prove that in this case the Poincaré modes can only exist in an exponentially decaying (or bound) form regardless of the basin width. In fact, if $f > \sigma$, we then have

$$\alpha^2 - \beta^2 = [(\frac{f}{\sigma})^2 - 1]k^2 > 0 \tag{R7}$$

This implies $s_n > 1$ according to Eq. (R4) above (in the above derivation we have neglected friction for simplicity as did in deriving (R1)).

Remarks:

1) "- I do not understand the distinction on tides vs tide dynamics made by the authors when rebutting my request to expand on earlier literature on the study site."

Reply: In our opinion this is similar to the case of physical oceanography versus dynamical physical oceanography: the former includes the latter and another discipline, descriptive physical oceanography. The review on earlier studies has been expanded in R2, please see line 11, p. 5 to line 11, p. 6 of TPR below.

2) "- I somewhat regret that the theoretical effort of deriving a frictional radiating condition is not taken, even though its effect is shown to be small. I personally think that adopting the idealized modelling philosophy implies that - once the model has been formulated - one should try to avoid unnecessary approximations in the mathematical solution as much as possible."

Reply: According to this comment, we have changed the radiative condition with friction included in R2. That is, Eq. (6) is changed from $u = \pm\sqrt{\frac{g}{h}}\zeta$ in R1 to $u = \pm\sqrt{\frac{g}{(1-i\mu)h}}\zeta$ in R2. Correspondingly the statement "Strictly speaking, Eq. (6) is valid only in the frictionless case; if friction is considered, it contains an error of the order of

$\mu^2$, and there is a phase difference between $u$ and $\zeta$ (Fang and Wang, 1966). In the present study we still use the form of Eq. (6) due to smallness of the value $\mu$" in R1 now is replaced with the statement "The relationship between $u$ and $\zeta$ shown in Eq. (6) is based on the solution for progressive Kelvin waves in presence of friction, which will be given in Eqs. (9) and (10) below. If $\mu \ll 1$, a simpler equation $u = \pm\sqrt{\frac{g}{h}}\zeta$ can be used to replace Eq. (6)" in R2.

Experiments 1-3 in the subsection 4.1 (which involve the radiative condition) have been re-computed. Figures 5-7 are also replaced with the re-computed results. The differences between the original and re-computed results are very small. As an example, a comparison between the corresponding results for Ex. 1 is shown below. From the figure we can see that the differences are almost invisible, except that in Area B, where the amplitude of northward Kelvin wave reduces from about $6 \times 10^{-4}$ m to about $3 \times 10^{-16}$ m. The latter is more reasonable, though both are very close to zero.

[Figure]

[Figure]

Figure R1. Comparison between the results using different radiative conditions for Ex.1. The upper panel is the result using radiative condition $u = \sqrt{\frac{g}{h}}\zeta$; the lower using $u = \sqrt{\frac{g}{(1-i\mu)h}}\zeta$.

Please do make the correction.

Regarding remark 1, the point being made by the referee was that you list (page 2, lines 11-13) many references on tides in Taiwan Strait without saying something about what is learned from those references. There should be some benefit to the paper in citing each reference.

Reply: According to this comment the statement "Most studies have attempted to establish accurate numerical models and thus to give accurate spatial structures of the tides and tidal currents in the strait (Yin and Chen, 1982; Fang et al., 1984; Ye et al., 1985; Lü et al., 1999; Lin et al., 2000; Lin et al., 2001; Jan et al., 2002; Jan et al., 2004a; Jan et al., 2004b; Zhu, et al., 2009; Hu et al., 2010; Zeng et al., 2012; Yu et al., 2015; Yu et al., 2017)" has been replaced with a more detailed description: "Yin and Chen (1982) first developed two-dimensional model for tides in the TS without showing tidal

currents. Fang et al. (1984) again developed a two-dimensional model and obtained rather accurate distribution of tidal currents. They suggested that the semidiurnal tidal motion in the TS was maintained mainly by the energy flux from the ECS and partly by that from the SCS. Ye et al. (1985) and Lü and Sha (1999) developed three-dimensional models for the strait and also found that the southward energy flux of semidiurnal tides from the ECS was much greater than the northward flux. Lin et al. (2000, 2001) emphasized the anomalous amplification of semidiurnal tides in the strait, and attributed the amplification to a resonance. Jan et al. (2004b) modeled the tides using an optimization approach. Zhu, et al. (2009), Hu et al. (2010), and Zeng et al. (2012) further developed more accurate numerical models. Yu et al. (2017) studied the propagation and dissipation of tidal waves in the strait".

Regarding remark 2, I think the effect of friction as represented in your manuscript is to replace "h" in (6) by "h(1-iμ)" (making the same simplification as for (6) which applies to a Kelvin wave with v=0).

Reply: Yes, we have made corresponding change. Please see line 26, p.3 to line 4, p. 5 of TPR.

Additional Editor comments.

Page 5 lines 17-18. I think "if friction is considered, it contains an error of the order of μ2," refers to the amplitude (only; not the phase).

Reply: In R2 friction has been included in the radiative condition, the statement "if friction is considered, it contains an error of the order of $\mu^2$" now is not needed. Please see reply to referee' comment above (line 26, p. 3 to line 6, p. 4 of TPR).

Page 7 lines 3-4. "the high-order Poincaré modes decay from the boundary very quickly (e.g., Godin, 1965)"; this can be seen directly from large sn in (11), (12).

Reply: "(e.g., Godin, 1965)" has been replaced with "as can be seen from large $s_n$ in (11) and (12)".

Page 8 line 11. "northeast" -> "northwest"?

Reply: Here we describe the wave propagation along the Taiwan coast. For clarity we have replaced the statement "the wave propagates southward in the northeast area and propagates northward in the southeast area" with the statement "along the Taiwan coast the wave propagates southward in the northern area and propagates northward in the southern area".

Page 10 line 5. "northward Kelvin wave is weaker than the northward wave on average". Do you mean ". . weaker than the southward wave . ."?

Reply: Yes, the second "northward" is wrong, in R2 it has been replaced with "southward".

Additional Note: The following are your comments we received on Nov. 7, 2017. In R2 these suggestions have all been adopted and the text is revised accordingly. We copy these comments below.

Meanwhile I have a few editorial comments for you to take into account in any final version.

Lines 11-12.   Better ". . The largest amplitudes are roughly along the cross section of 150 km and appear as an anti-nodal band. . . ."
Lines 17-18.   Better ". . using only superposed Kelvin waves (Fig. 3c) . ."

Line 1.   Better ". . Poincaré modes play a role . ."
Line 2.   Better ". . modes make nearly . ."
Line 8.   Better ". . and superposed Poincaré modes are necessary . ."
Line 9.   ". . Poincaré modes in . ."
Line 11.   ". . but has an angle . ." (this is still true!)
Line 24.   Better ". . tides here have the . ."
Lines 24-25.   ". . two waves, exceeding 2.1 m, as already seen . ."

Page 10.
Line 5.   ". . northward Kelvin wave is weaker than the southward wave on average . ."?
Bottom line.   Better ". .   there is again an anti-nodal band . ."

Page 12 line 6.  ". . performed a second . ."

Page 13 line 4.  ". . performing a third . ."

Page 14 line 12.  ". . Kelvin wave, with . ."

**Modification list**
(According to Revision#2)

**Page 2.**
Lines 14-23. The statement "Most studies have attempted to establish accurate numerical models and thus to give accurate spatial structures of the tides and tidal currents in the strait (Yin and Chen, 1982; Fang et al., 1984; Ye et al., 1985; Lü et al., 1999; Lin et al., 2000; Lin et al., 2001; Jan et al., 2002; Jan et al., 2004a; Jan et al., 2004b; Zhu, et al., 2009; Hu et al., 2010; Zeng et al., 2012; Yu et al., 2015; Yu et al., 2017)" has been replaced with a more detailed description:

"Yin and Chen (1982) first developed two-dimensional model for tides in the TS without showing tidal currents. Fang et al. (1984) again developed a two-dimensional model and obtained rather accurate distribution of tidal currents. They suggested that the semidiurnal tidal motion in the TS was maintained mainly by the energy flux from the ECS and partly by that from the SCS. Ye et al. (1985) and Lü and Sha (1999) developed three-dimensional models for the strait and also found that the southward energy flux of semidiurnal tides from the ECS was much greater than the northward flux. Lin et al. (2000, 2001) emphasized the anomalous amplification of semidiurnal tides in the strait, and attributed the amplification to a resonance. Jan et al. (2004b) modeled the tides using an optimization approach. Zhu, et al. (2009), Hu et al. (2010), and Zeng et al. (2012) further developed more accurate numerical models. Yu et al. (2017) studied the propagation and dissipation of tidal waves in the strait".

Line 24. "and Ye et al. (1985)" has been added after "Fang et al. (1984, 1999)".

Line 26. "2004" has been changed to "2004a".

**Page 5.**

Line 2. "$u = \sqrt{\frac{g}{h}}\zeta$" has been changed to "$u = \pm\sqrt{\frac{g}{(1-i\mu)h}}\zeta$".

**Page 6.**
Lines 3-5. The statement "Strictly speaking, Eq. (6) is valid only in the frictionless case; if friction is considered, it contains an error of the order of $\mu^2$, and there is a phase difference between $u$ and $\zeta$ (Fang and Wang, 1966). In the present study we still use the form of Eq. (6) due to smallness of the value $\mu$." has been changed to "The relationship between $u$ and $\zeta$ shown in Eq. (6) is based on the solution for progressive Kelvin waves in presence of friction, which will be given in Eqs. (9) and (10) below. If $\mu \ll 1$, a simpler equation $u = \pm\sqrt{\frac{g}{h}}\zeta$ can be used to replace Eq. (6)."

Line 19. The equation (16) "$s_n = (r_n^2 - \alpha^2 + \beta^2)^{\frac{1}{2}}$" has been changed to "$s_n = (r_n^2 + \alpha^2 - \beta^2)^{\frac{1}{2}}$".

**Page 7.**
Line 17. "(e.g. Godin 1965)" has been changed to "as can be seen from large $s_n$ in (11) and (12)".

**Page 8.**
Lines18-19. "along the Taiwan coast" has been added before "the wave propagates southward" and the words "northeast" and "southeast" have been changed to "northern" and "southern", respectively.

Line 19. "The highest amplitude band presents roughly…" has been changed to "The largest amplitudes are roughly…".

Line 25. The word "a" has been replaced by "only".

**Page 9.**

Line 1. "wave alone" has been changed to "waves".

Line 9. "Poincaré mode plays" has been changed to "Poincaré modes play".

Line 10. "Poincaré mode has" has been changed to "Poincaré modes make".

Line 16. "Poincaré mode is" has been changed to "Poincaré modes are".

Line 17. "Poincaré mode" has been changed to "Poincaré modes".

Line 19. ".. but had an angle" has been changed to "… but has an angle".

Line 20. The word "here" has been added after "… the superposed tides".

**Page 10.**

Line 1. "which exceeds" has been changed to "exceeding".

Line 20. "… is weaker than the northward wave …" has been changed to "… is weaker than the southward wave …".

**Page 11.**

Line 4. "… there is also an anti-nodal band …" has been changed to "… there is again an anti-nodal band …".

**Page 12.**

Figure 5. (and other places: Page 13. Figure 6 and Page 14. Figure 7) has been replaced with a re-computed results.

Line 6. "… we performed the second experiment …" has been changed to "… we performed a second experiment …"

**Page 13.**

Line 4. "… performing the third experiment …" has been changed to "… performing a third experiment …"

**Page 14.**

Lines 12. "Kelvin waves" have been changed to "Kelvin wave".

**Revision #2**

[revised manuscript text omitted]